# Group A *Streptococcal* asparagine metabolism regulates bacterial virulence

Abhinay Sharma [1,5], Aparna Anand[1,5], Miriam Ravins[1], Xiaolan Zhang [2], Nicola Horstmann [3], Samuel A Shelburne[3], Kevin S McIver [4] & Emanuel Hanski [1✉]

## Abstract

**Group A Streptococcus (GAS) causes various human diseases linked to virulome expression predominantly regulated by the two-component system (TCS), CovR/S. Here, we demonstrate that asparagine (Asn) presence in a minimal chemically defined medium increases virulence gene expression in a CovR-dependent fashion. It also decreases the transcription of asparagine synthetase (AsnA), the ABC transporter responsible for Asn uptake (GlnPQ), and that of the hemolysin toxins responsible for scavenging Asn from the host. Metabolomics data show that Asn availability increases intracellular ADP/ATP ratio, which enhances phosphatase activity in structurally related CovS sensors and is probably responsible for the Asn-mediated decrease in CovR phosphorylation. Mutants deficient in AsnA, GlnPQ, asparaginase, (AsnB) activities are attenuated in a mouse model of human GAS invasive soft tissue infection. The similarity between the mechanisms of Asn-mediated regulation of GAS virulence and tumor growth suggests that, as in cancer, components maintaining Asn homeostasis could be targeted for anti-GAS treatments.**

**Keywords** Group A Streptococcus; Metabolism; Asparagine; Virulence; Regulation Mechanism
**Subject Categories** Metabolism; Microbiology, Virology & Host Pathogen Interaction; Signal Transduction

## Introduction

*Streptococcus pyogenes*—Group A streptococcus (GAS) is an extracellular strict human pathogen recognized among the top ten causes of infection mortality (Brouwer et al, 2023). GAS causes many human infections, including life-threatening ones, such as bacteremia, necrotizing fasciitis (NF), and streptococcal toxic shock syndrome STSS (Brouwer et al, 2023; Cole et al, 2011). In addition, GAS may trigger a lethal autoimmune sequela (Cunningham, 2019;

Martin et al, 2015). Between 2022 and 2024, a surge in invasive GAS diseases in Europe, the USA, and Japan was observed (Bagcchi, 2023; Freiberg and Wright, 2024; Kim, 2024; Nygaard et al, 2024).

Bacterial pathogens respond to specific nutritional cues within host microenvironments, and growth within these microenvironments requires metabolic adaptation. Studies along these lines peaked in recent years, and novel traits have been uncovered mostly for intracellular pathogens or pathogens occupying specific host niches (Brown et al, 2008). However, we know little about extracellular pathogens, such as GAS, that co-exist with their host, cause many diseases, and pass-through various host environments during the infectious process (Brouwer et al, 2023). We demonstrated that the acquisition of Asn from the host by GAS during infection is a critical trait in GAS pathogenesis (Baruch et al, 2014). GAS delivers the toxins streptolysin S (SLS) and streptolysin O, activating the PERK-eIF2α-ATF4 branch of the unfolded protein response (UPR), resulting in the upregulation of host asparagine synthetase (ASNS) transcription (Anand et al, 2021; Gjymishka et al, 2009). Consequently, the Asn level is augmented in infected host cells (Anand et al, 2021; Baruch et al, 2014). By inhibiting the PERK-eIF2α-ATF4 pathway using specific inhibitors, we protected mice against invasive GAS diseases in murine models of human soft tissue GAS infection (Anand et al, 2021).

While the mechanism by which GAS scavenges Asn from the host is well studied, little is known about how Asn regulates the virulence of GAS. Although GAS is a multiple-amino-acid-auxotrophic bacteria (Davies et al, 1965), it can synthesize Asn by asparagine synthase (AsnA) from aspartic acid and ammonia while consuming ATP. It also possesses asparaginases (AsnB), degrading Asn to aspartate and ammonia. The transporters responsible for the uptake of scavenged Asn from the host have not yet been identified. Several ABC-transporters whose transcription is downregulated by Asn presence have been implicated (Baruch et al, 2014), including the ATP-binding cassette (ABC) importer GlnPQ, which possesses two substrate-binding domains of high affinity towards arginine, Asn glutamine, and glutamate (Fulyani et al, 2013; Gouridis et al, 2015; Nemchinova et al, 2024). In *Listeria monocytogenes*, GlnPQ is the only high-affinity import system responsible for the uptake of L-glutamine (Haber et al, 2017).

[1]Department of Microbiology and Molecular Genetics, The Institute for Medical Research, Israel-Canada (IMRIC), Faculty of Medicine, The Hebrew University of Jerusalem, Jerusalem 9112102, Israel. [2]Department of Physiology, College of Basic Medical Science, Harbin Medical University, Harbin, China. [3]Department of Infectious Diseases, Infection Control and Employee Health, MD Anderson Cancer Center, Houston, TX, USA. [4]Department of Cell Biology and Molecular Genetics, Maryland Pathogen Research Institute, University of Maryland, College Park, MD 20742, USA. [5]These authors contributed equally: Abhinay Sharma, Aparna Anand. ✉E-mail: emanuelh@ekmd.huji.ac.il

Control of virulence (CovR/S) or capsule synthesis regulator (CsrR/S) is the best-characterized two-component system (TCS), which is a primary regulator of GAS virulence (Graham et al, 2002; Vega et al, 2022). CovS shares domain organization and structural similarities with EnvZ and PhoQ sensors of a two-component signal transduction family possessing histidine kinase (HK) and phosphatase activities (Jacob-Dubuisson et al, 2018; Krell et al, 2010). ATP is a substrate for HK, whereas ADP stimulates its phosphatase activity (Castelli et al, 2000; Igo et al, 1989; Sanowar and Le Moual, 2005; Zhu and Inouye, 2002; Zhu et al, 2000). When phosphorylated, CovR represses virulence factor transcription (Vega et al, 2022). Cathelicidin host-defense peptide (LL-37) binds to CovS and increases its phosphatase activity, thereby enhancing virulence gene transcription (Horstmann et al, 2018; Velarde et al, 2014).

Here, we show how supplementing a chemically defined medium (CDM) with Asn affects the transcriptome in GAS. We observed the upregulation of various CovR/S controlled genes, including virulence factors, while some virulence genes, such as streptolysin toxins (SLO and SLS), were downregulated by Asn independently of CovR. Furthermore, we found that GlnPQ is solely responsible for importing Asn. We further demonstrated that either AsnA, GlnPQ, or AsnB mutants are avirulent in the murine model of GAS human NF. Metabolic analysis shows that Asn availability decreases intracellular ATP, leading to a high ADP/ATP ratio. As ADP stimulates the phosphatase activity of sensors structurally related to CovS, we propose that Asn regulates GAS virulence by altering the ADP to ATP concentrations. In addition, our study proposes a similarity between the mechanism of Asn-mediated regulation of GAS virulence and tumor growth, suggesting, as in cancer, the therapeutic potential of AsnA and AsnB as drug targets against GAS.

# Results

## Characterization of Asn-mediated regulatory circuits dependent and independent of CovR/S by RNA-seq and qPCR

To decipher the effect of Asn on GAS gene transcription, we performed transcriptome sequencing (RNA-seq) experiments in CDM. Supplementing the CDM with Asn at increasing concentrations of 0.5–10 μg/ml enhanced the S119 growth kinetics, with faster growth observed at 10 μg/ml (Fig. 1A). Nevertheless, after prolonged growth (overnight), the cultures with and without Asn reached similar optical density (Fig. 1A), indicating that GAS AsnA might be sluggish. We found that GAS grew somewhat faster in 100 μg/ml than 10 μg/ml Asn, but the increase in growth was insignificant.

We compared the transcriptome of S119 grown to an $OD_{600} = 0.7$ in CDM supplemented with Asn (10 μg/ml) or not using RNA-seq experiments. Genes with at least a twofold increase or decrease in transcript abundance have been summarized in a heatmap (Fig. 1B). These genes belong to functional categories like Cell communications, Cell division/replication proteins, Hypothetical proteins, Kinases, Metabolic enzymes, Phage-associated proteins, Regulatory proteins, Stress related proteins, Transferases, Transporters/ABC transporters/solute binding proteins, and Virulence factors (Fig. 1B). The heatmap shown in Fig. EV1A represents the expression profile of several genes including virulence factors regulated by CovR directly or indirectly; the raw data for both

heatmaps is provided in GEO accession number GSE268517. Out of the total number of detected genes ($n = 1839$), 27.35% ($n = 503$) were affected; out of these genes, 13.97% ($n = 257$) were upregulated, and 13.37% ($n = 246$) were downregulated by at least twofold in the treatment condition ( + Asn). The most upregulated genes were SHP2 and SHP3, encoding autoinducing peptide pheromones of the quorum sensing Rgg2/3 pathways leading to increased biofilm formation (Gogos et al, 2018). In addition, glucose-6 phosphate isomerase (a highly conserved glycolytic enzyme), glycosyltransferase (involved in cell wall biogenesis), and the eukaryotic-type serine/threonine kinase (involved in cell cycle regulation) (Mikkat et al, 2024) were also found to be highly upregulated (Fig. EV1B,C). The strongest downregulated genes belonged to unknown function categories (Fig. EV1B,C).

RNAseq results were validated by performing quantitative reverse transcription PCR (qRT-PCR) determinations of a few upregulated (emm, scpA, scpC, ska, hasA) and downregulated genes (slo, sagA, asnA, and glnPQ) (Fig. 1C,D). AsnA transcription was downregulated in Asn presence (Fig. 1D), suggesting that Asn acts as an allosteric feedback inhibitor of AsnA (Norton and Chen, 1969). However, the transcription of the genes encoding SLO and SLS, both known to be derepressed by unphosphorylated CovR (Friaes et al, 2015; Langshaw et al, 2023), was repressed under these conditions (Fig. 1D).

Since the transcription of glnPQ was upregulated upon adding the Gram-negative asparaginase, Kidrolase, into a semi-rich medium of the M14 strain JS95 (Baruch et al, 2014), we tested if Asn would affect glnPQ transcription of strain S119 in the minimal CDM. The qRT-PCR results show that glnP transcript was significantly more abundant when S119 was cultured in the absence of Asn than in its presence ($P = 0.0003$) (Fig. 1D), suggesting that GlnPQ could participate in the regulatory circuit of SLO, SLS, and AsnA and thus be responsible for Asn import.

Since most of the virulence genes are known to be regulated directly and indirectly by CovR/S (Finn et al, 2021; Horstmann et al, 2022; Horstmann et al, 2018), we aimed to examine whether CovS and CovR play any role in the regulation of these genes by Asn. We used S119 mutants with precise deletion of CovS that expresses CovR and insertional inactivation of CovR, which prevents the expression of both CovS and CovR, and determined the transcription pattern of several genes upregulated by Asn (Set 1) and downregulated (Set 2). In the CovS-deficient S119 mutant, Set 1 genes were upregulated, and Set 2 genes were downregulated in CDM supplemented with Asn except for slo and sagA (Fig. 1E,F). However, when covR and covS were inactivated, Set 1 genes were no longer upregulated by Asn. In contrast, the Asn-mediated negative regulation of glnP and asnA was still maintained (Fig. 1G,H). This finding implies that CovS does not directly affect Asn-mediated gene regulation, but CovR does, except for glnP and asnA, which are inhibited by Asn independently of CovR. Interestingly, SLO and SLS expression, dependent on CovR, was negatively regulated by Asn, possibly connecting between Asn homeostasis and virulence.

## Characterization of the control of S119 gene transcription by AsnA, GlnPQ and AsnB

### Control by AsnA

To evaluate the role of asparagine synthetase (AsnA) in GAS growth and gene transcription, we constructed a deletion mutant in asnA

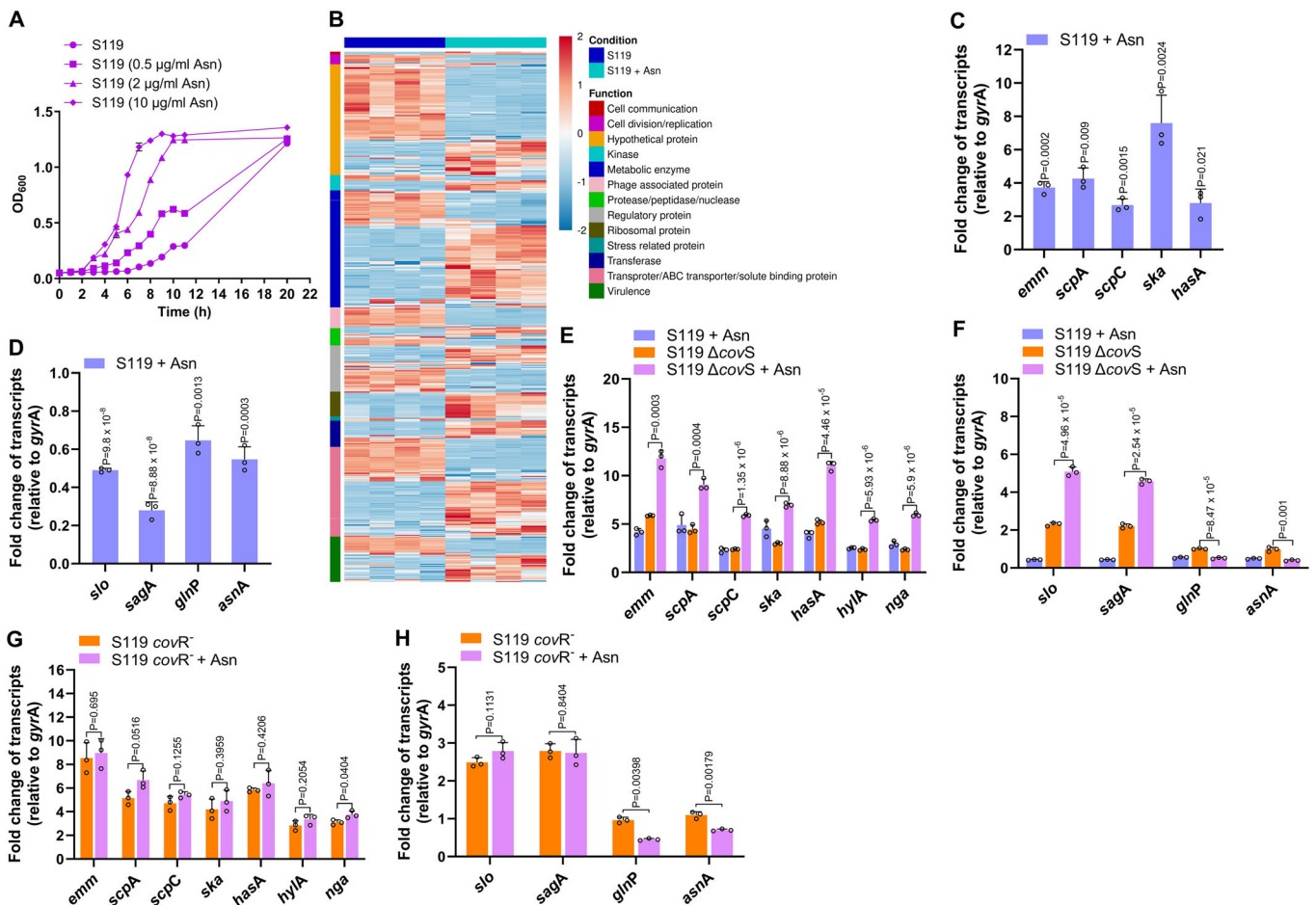

**Figure 1. Asn affects GAS growth and transcription.**

(A) The growth of the GAS strain S119 was determined in CDM in the absence or presence of Asn (0.5, 2, 10 µg ml⁻¹). (B) The heatmap shows differential gene expression patterns based on RNA-seq data. Data illustrates the global differential expression of genes belonging to the indicated different functional categories. (C, D) Quantitative real-time PCR (qRT-PCR) validations of mRNA-seq data were performed. Upregulated (emm, scpA, scpC, ska, and hasA) (C) and downregulated genes (slo, sagA, glnP, and asnA) (D) are presented. The determinations were performed on the three RNA samples used for the RNAseq experiment. (E–H) qRT-PCR determinations of Asn effect on the transcription of selected genes (emm, scpA, scpC, ska, hasA, hylA, nga) Set 1 (E, G), (slo, and sagA, asnA, glnP) Set 2 (F, H), in S119 and its derived ΔcovS (E, F) and covR⁻ (G, H) mutants in CDM without or with Asn (10 µg ml⁻¹). In all qRT-PCR data, transcript abundance for each gene was normalized to that of gyrA in each sample, and fold change was calculated in comparison with the normalized transcript abundance of the S119 grown without Asn (C–H). Data information: Three (A, C–H) and four (B) biological replicates were used. The values shown represent the means ± SD. Statistical analysis was performed using an unpaired two-tailed t test (C–H). Source data are available online for this figure.

(ΔasnA). We found that ΔasnA is Asn-auxotrophic when cultured in CDM (Fig. 2A). Its maximum growth could be restored by supplementing the CDM with Asn or Ala-Asn dipeptide (Fig. 2A), suggesting that Asn cannot be formed by ΔasnA when cultured in CDM, but extracellular Asn or Ala-Asn supports growth when brought in via Asn importer or likely through the dipeptide permease (Podbielski and Leonard, 1998). In addition, the increase in the growth kinetics of the AsnA-deficient mutant was restored in CDM without Asn by genetic complementation with a WT asnA gene expressed from a plasmid (Fig. EV2A). A plasmid expressing a mutated AsnA containing a single amino acid mutation that rendered it catalytically inactive (replacement of arginine with lysine at amino acid 100, Arg100Lys) did not complement the growth without Asn (Fig. EV2A).

Next, we found that the ΔasnA mutant lost the Asn-mediated upregulation of genes belonging to Set 1 using qRT-PCR (Fig. 2B).

However, it was restored by complementing the mutant with a plasmid expressing the WT gene but not with an Arg100Lys catalytically inactive allele, suggesting the formation of Asn is essential for the AsnA-mediated regulation (Fig. EV2B). Furthermore, the negative feedback regulation controlling the expression of the genes encoding SLO and SLS was lost in the ΔasnA mutant and restored by complementation with the WT gene and not by the catalytically inactive gene (Fig. EV2C). Interestingly, in the ΔasnA mutant, the transcription of glnP was increased by about sixfold (Fig. 2C). This likely compensated for the loss of Asn production, suggesting that glnPQ and asnA regulation are tightly linked. In addition, we demonstrated that the deletion of asnA in the M1T1 strain 854 exerted similar effects on its growth in CDM and gene regulation (Fig. EV2D–F). The role of RocA, known to participate in CovR regulation, was ruled out in these processes (Fig. EV2G–I).

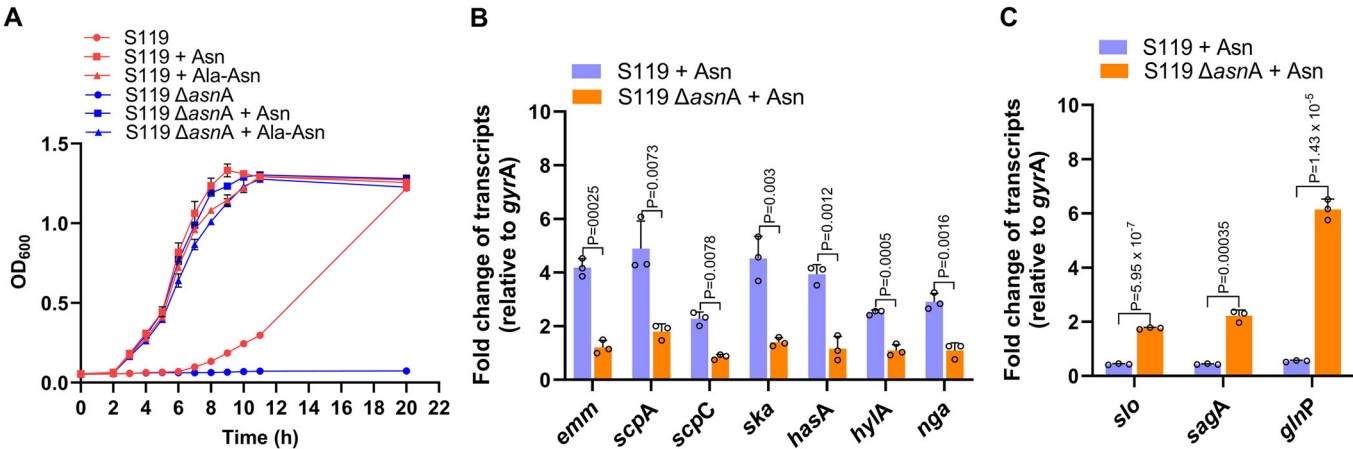

**Figure 2. *asn*A is essential for Asn-mediated gene regulation.**

(A) The strains S119 and S119Δ*asn*A were grown in CDM in the absence or presence of Asn (10 µg ml⁻¹) or dipeptide (Ala-Asn) (100 µg ml⁻¹), and $OD_{600}$ was determined at indicated time intervals. (B, C) qRT-PCR determinations of Set 1 (B) and Set 2 (C) of S119 and S119 Δ*asn*A genes were performed as in Fig. 1. (B, C) Data information: Three biological replicates were used (A–C). The data shown represent the means ± SD. Statistical analysis was performed using an unpaired two-tailed *t* test (B, C). Source data are available online for this figure.

In summary, catalytically active AsnA is necessary for Asn-mediated gene regulation but not for growth in the CDM supplemented with Asn or dipeptide-containing Asn.

### Control by GlnPQ

To evaluate the role of the ATP-binding cassette transporter GlnPQ on GAS growth and gene transcription, we constructed an insertion-inactivation mutant (*gln*P⁻) and monitored its growth pattern in CDM or CDM supplemented with Asn. The *gln*P⁻ and WT strain S119 grew in the absence of Asn at similar slow rates, but the *gln*P⁻ mutant did not increase its growth kinetics in CDM supplemented with 10 or even 100 µg/ml of Asn (Fig. 3A). Genetic complementation with the WT *gln*P gene expressed from a plasmid restored the enhanced growth phenotype of the mutant in the presence of Asn (Fig. 3A). These results show that GlnPQ is the main importer of Asn in GAS.

To provide direct evidence that GlnPQ is responsible for the uptake of Asn in GAS, we grew S119 WT or *gln*P⁻ to $OD_{600} = 0.2$ in CDM without Asn. Then, Asn (10 µg/ml) was added or not, and we followed the growth of both strains. As expected from Fig. 3A, adding Asn enhanced the growth kinetics of S119 but not of *gln*P⁻, which maintained a relatively slow growth kinetics despite the presence of Asn (Fig. EV3A). To follow the uptake of Asn, we quantified the Asn concentration in the media at different time points of the two strains by liquid chromatography-mass spectrometry analysis (LC-MS) (Mackay et al, 2015). We found that most of the Asn was taken up from the media by S119 within 60 min. In contrast, the Asn concentration in the media of the *gln*P⁻ did not change at all and was equal to that found in fresh media (Fig. 3B). This experiment demonstrates that GlnPQ is solely responsible for Asn uptake by GAS under our experimental conditions.

To assess how the deficiency in GlnPQ activity affects gene regulation, we repeated the qRT-PCR determination described above. We found that the upregulation in the transcription of gene Set 1 was lost entirely in the *gln*P⁻ mutant in the presence of Asn (Fig. 3C). The negative feedback regulation of Set 2 was also lost (Fig. 3D). Interestingly, the *asn*A expression in CDM both in the absence and presence of Asn was similar and about twofold higher than that of the WT strain in the presence of Asn (Fig. 3D), presumably to compensate for the deficiency in the uptake of Asn from the medium (Fig. 2C). Finally, expressing WT *gln*P from a plasmid restored the WT virulence gene regulation (Fig. EV3B), *slo*, *sag* (SLS), *asn*A, and *gln*PQ genes (Fig. EV3C), confirming the role of *gln*PQ in the regulation.

### Control by AsnB

Although the transcription of GAS asparaginase (AsnB) was not affected by Asn's absence or presence in CDM or by deficiency of AsnA or GlnPQ activity (Fig. EV3D), we decided to construct an *asn*B⁻ mutant and examine its effect on GAS growth and gene regulation. The main reason was that AsnB belongs to the type II asparaginase family, having kcat and $K_M$ ranging around 12–60 s⁻¹ and 10–20 µM, respectively (Lubkowski and Wlodawer, 2021). Therefore, AsnB should leverage intracellular Asn concentrations due to its high enzymatic efficiency. We measured L-asparaginase activity in bacterial-cell suspensions to confirm the expected *asn*B⁻ phenotype (Farahat et al, 2020). Asparaginase activity of *asn*B⁻ was lost but restored when the mutant was complemented by the WT gene expressed from a plasmid ($P = 0.1648$) (Fig. EV3E). The deficiency in asparaginase activity increased the GAS growth kinetics in CDM in the absence of Asn (Fig. EV3F). However, the transcription of Set 1 of genes in *asn*B⁻ was not upregulated when CDM was supplemented by Asn compared to S119 supplemented with Asn (Fig. 3E). The transcription of the genes encoding SLO, SLS, AsnA, and GlnPQ was reduced compared to the WT S119 was slightly further reduced by Asn presence (Fig. 3F). Furthermore, upon genetic complementation with a WT gene expressed from a plasmid, the Asn-mediated regulation of genes belonging to Set 1 and the genes encoding SLO, SLS, and AsnA was regained (Fig. EV3G,H). These findings suggested that Asn-mediated transcription regulation occurs at a defined Asn intracellular concentration range. This range is affected by the interplay between the Asn intracellular concentration and its effect on the

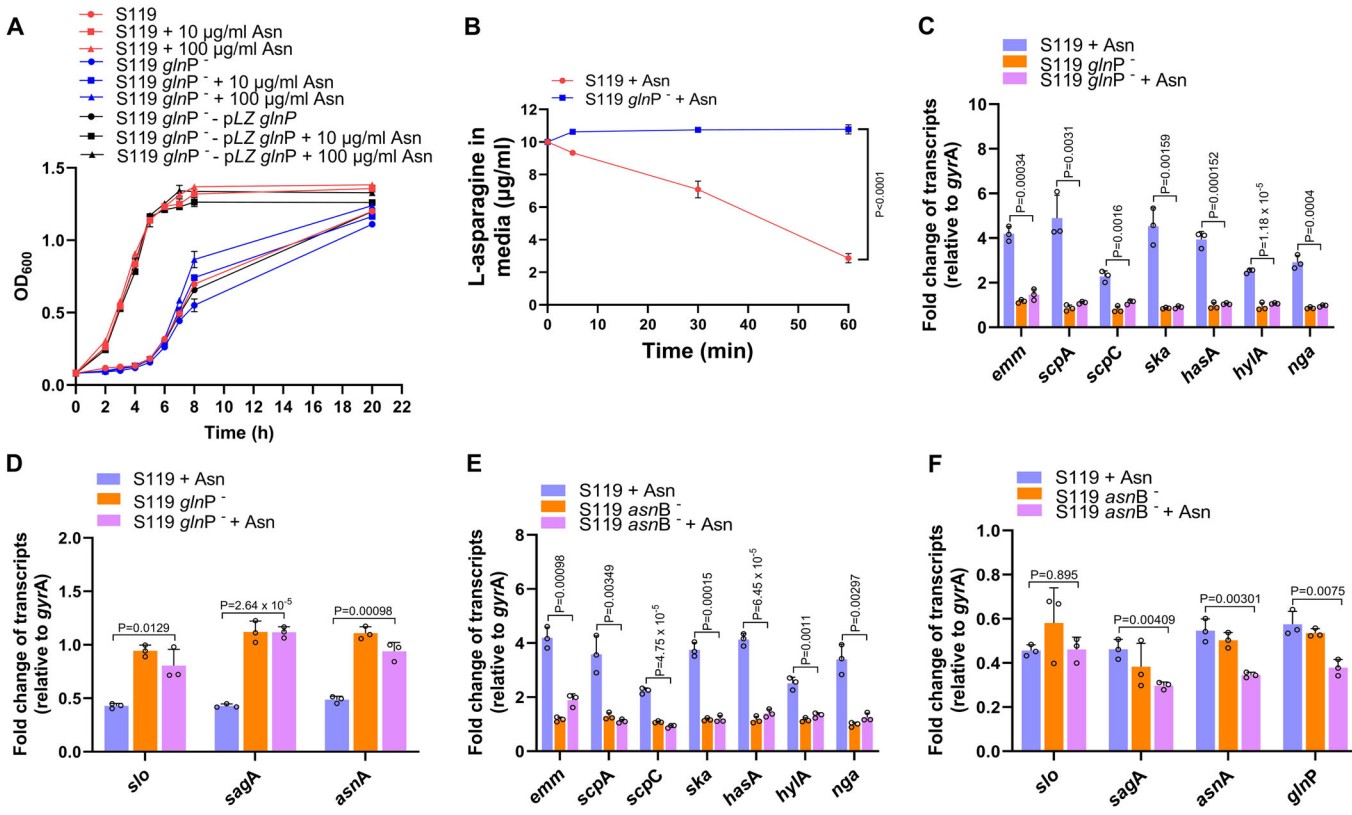

**Figure 3. glnP and asnB are essential for Asn-mediated gene regulation.**

(A) The growth of S119, S119 glnP⁻ and S119 glnP⁻-pLZglnP in CDM in the absence or presence of Asn (10 and 100 µg ml⁻¹) was determined at indicated time points. (B) S119 or its glnP⁻-derived mutant GAS was cultured in CDM without Asn to OD$_{600}$ = 0.2, and then Asn was added. The Asn concentration in the medium was determined by LC-MS at 5, 30, and 60 min after Asn addition. (C–F) qRT-PCR determinations were performed on Set 1 (C, E) and Set 2 of genes (D, F) comparing S119 and S119 glnP⁻ (C, D) and S119 and S119 asnB⁻ (E, F) grown in CDM without or with Asn. In all qRT-PCR data, transcript abundance for each gene was normalized to that of the GAS S119 strain without Asn (C–F). Data information: Three (A, C–F) and five (B) biological replicates were used. The values shown represent the means ± SD. Statistical analysis was performed using two-way ANOVA (B) unpaired two-tailed t test (C–F). Source data are available online for this figure.

transcription of asnA and glnPQ and the impact of the transcription of the latter genes on each other (Fig. EV3I).

## Asn increases virulence factor expression by reducing CovR phosphorylation

To corroborate that an Asn-mediated increase in the transcription of virulence factors also increases the expression of the related proteins, we tested the activities of ScpC (a.k.a SpyCEP) and ScpA encoding the CXC-chemokine serine protease and C5a peptidase, respectively (Edwards et al, 2005; Hidalgo-Grass et al, 2006; Lynskey et al, 2017). ScpC cleaves interleukin-8 (IL-8), and ScpA cleaves the complement component 5a (C5a); the two cleavage processes can be visualized on SDS-PAGE (Hidalgo-Grass et al, 2006; Lynskey et al, 2017). In addition, it has been shown that LL-37 binds to CovS and stimulates its phosphatase activity, thereby diminishing CovR phosphorylation and consequently increasing ScpC and ScpA expression (Finn et al, 2021; Horstmann et al, 2018). Therefore, we first tested if growing S119 in CDM in the absence or presence of Asn and or LL-37 would affect the activities of ScpC and ScpA. We found a low ScpC activity for S119 grown in the CDM only. In the presence of either Asn or LL-37, some

cleavage of IL-8 was visualized. However, considerable cleavage was apparent when S119 was grown in CDM containing both Asn and LL-37 (Fig. 4A). We repeated this experiment using the S119ΔcovS mutant and found some cleavage when cultured in CDM alone or CDM containing LL-37. In contrast, we detected a high amount of cleavage of IL-8 when S119ΔcovS was cultured in CDM containing either Asn (P = 0.00237) or Asn and LL-37 (P = 0.0011) (Fig. 4B). Comparable results were obtained for S119 and its derived ΔcovS mutant when tested for cleavage of C5a (Fig. EV4A,B).

To further substantiate these findings, we quantified IL-8 cleavage using an ELISA assay for IL-8. The results show that Asn stimulated ScpC activity production (Fig. 4C). As expected, the ΔcovS mutant produced a higher activity in the absence of Asn due to a basal increase in virulence gene expression. However, Asn-mediated ScpC increase in activity was still significant (P = 0.00237) (Fig. 4C). In complete agreement with the data presented in (Fig. EV1C,D), covR⁻ mutation fully activated ScpC activity, eliminating further increase by Asn (Fig. 4C). Furthermore, the ΔasnA glnP⁻ and asnB⁻ mutants lost or partially lost their ability to produce ScpC in the presence of Asn, which was restored by their genetic complementation with WT genes (Fig. 4D).

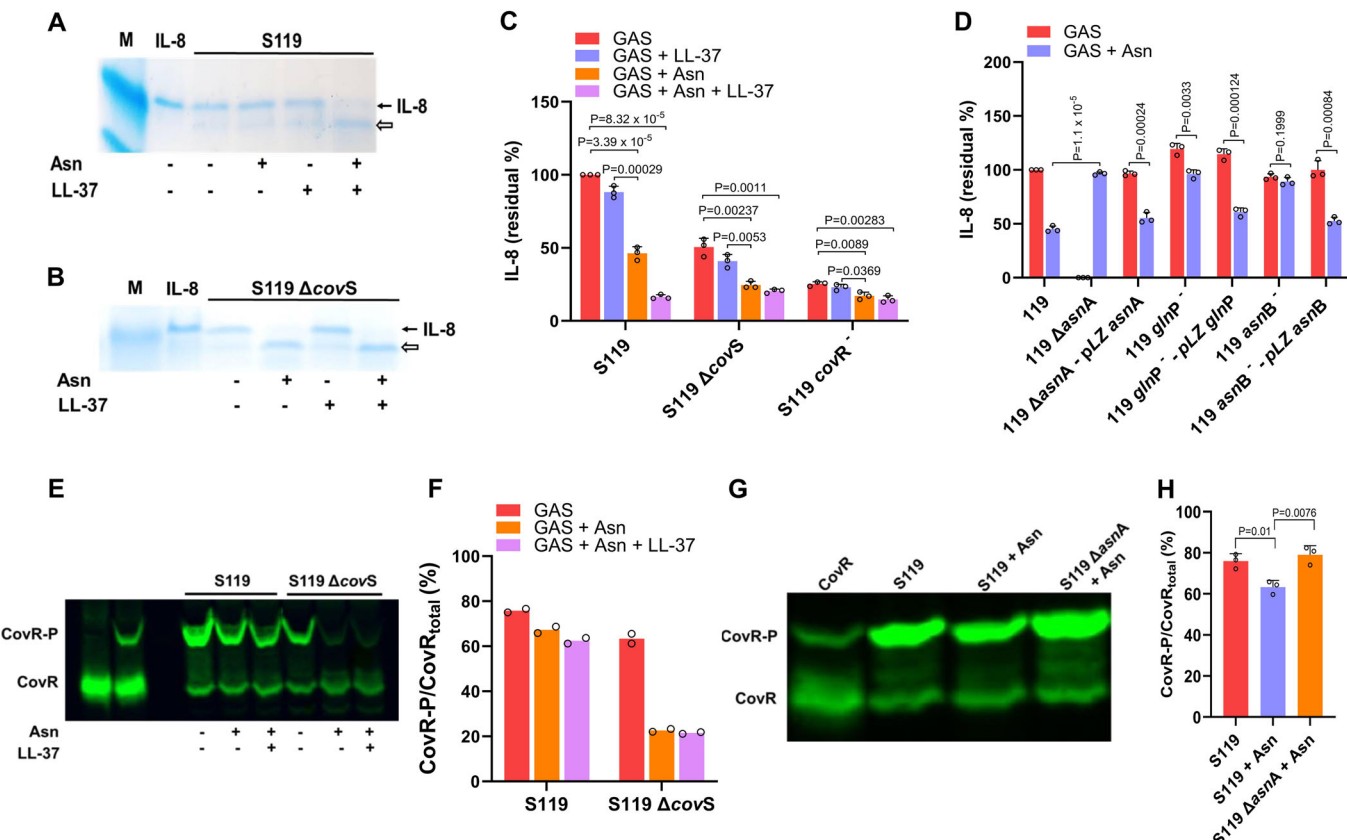

**Figure 4. Asn reduces the phosphorylation of CovR.**

(A, B) ScpC activity in culture media. Culture media of S119 (A), or its ΔcovS-derived mutant (B), were collected after growth in the absence or presence of Asn or/and LL-37 and then subjected to ScpC-mediated cleavage of recombinant human IL-8 followed by SDS-PAGE on Tris-tricine gels. The gels were visualized using Coomassie blue staining. (C, D) The determinations of IL-8 residual content in the supernatants of the indicated strains cultured without or with Asn or/and LL-37 were conducted by ELISA. The IL-8 residual content in all supernatants was normalized to that of the GAS S119 strain without Asn (C, D). (E, G) Asn reduced CovR phosphorylation. The indicated strains were grown in CDM without or with Asn or/and LL-37. Cell lysates (E, G) were separated by Phos-Tag SDS-PAGE, with unphosphorylated (lane 1, from left) and phosphorylated recombinant CovR protein (lane 2, from left) (E), CovR species were detected using an anti-CovR antibody and visualized using a fluorescently labeled secondary antibody (E, G). (F, H) The percentages of CovR-P of total CovR protein were calculated using ImageJ. Data information: Three (C, D, G, H) and two (E, F) biological replicates were used. The values shown represent the means ± SD. Statistical analysis was performed using an unpaired two-tailed $t$ test (B, D, H). Source data are available online for this figure.

---

Asn and LL-37 stimulated the activity of ScpC in the GAS 5448 strain (Fig. EV4C) and the GAS 854 strain ($P = 0.0001$) (Fig. EV4D). Moreover, the Asn-mediated stimulation of ScpC activity in the GAS 854 strain was abolished in its asnA⁻ mutant derivative ($P = 0.00012$), suggesting that the Asn-mediated upregulation is ubiquitous among GAS M1T1 strains (Fig. EV4C,D).

Since dephosphorylation of CovR upregulates the expression of GAS virulence factors (Finn et al, 2021; Horstmann et al, 2022; Horstmann et al, 2018), we tested if Asn-mediated activation would also affect CovR~P levels. To do so, we used Phos-Tag technology to quantify CovR phosphorylation (Horstmann et al, 2014). First, we established that Asn does not reduce the phosphorylation of purified CovR by acetyl phosphate in-vitro (Horstmann et al, 2015) (Fig. EV4E). Then, we assessed CovR phosphorylation during the growth of S119 in CDM in the presence of Asn compared to un-supplemented CDM. As shown in Fig. 4E, the presence of Asn reduced the level of phosphorylated CovR, and further reduction occurred in the presence of LL-37. Furthermore, Asn presence decreased the phosphorylation level of CovR in the S119ΔcovS

mutant by more than threefold (Fig. 4E,F). To validate that the Asn-mediated decrease in phosphorylation of CovR was abolished in the ΔasnA mutant, we compared the phosphorylation of CovR of ΔasnA and the WT in the presence of Asn (Fig. 4G). The deletion of AsnA prevented the Asn-mediated decrease in CovR phosphorylation ($P = 0.0076$) (Fig. 4G,H).

## Mutants in AsnA, GlnPQ, and AsnB are attenuated in a murine model of human NF

To test whether the mutants of AsnA, GlnPQ, and AsnB would have attenuated virulence during infection, we subjected ΔasnA, glnP⁻, and asnB⁻ to the sublethal soft tissue murine model of human NF that mimics the pathophysiology of the infection (Ravins et al, 2022). We enumerated colony-forming units (CFU) in soft tissue and spleen at different time intervals of 2, 4, 6, and 8 days after infection (Fig. 5A–F). We also pictured the size of the developed lesions (Fig. EV5A–C) and measured the lesion area (Fig. 5G–I). In addition, we determined the spleen weights

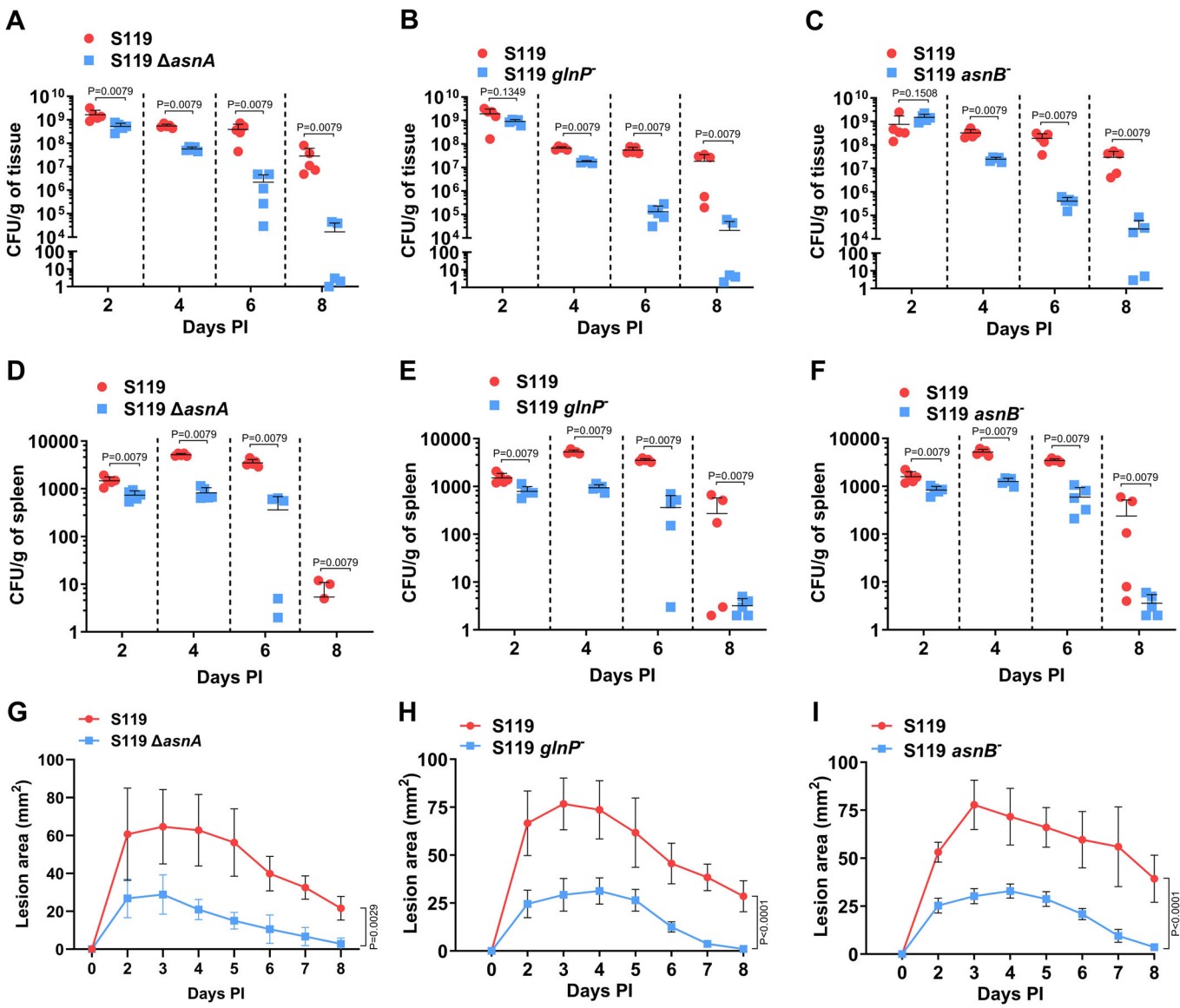

**Figure 5.** *asn*A, *gln*P, and *asn*B mutants are attenuated in the sublethal murine model of human NF.

(A–C) BALB/c mice were injected with a sub-lethal dose of GAS through a subcutaneous (SC) route. CFU counts per gram of soft tissue derived from mice infected with S119 Δ*asn*A (A), S119 *gln*P⁻ (B), and S119 *asn*B⁻ (C) compared to the wild-type bacteria S119 were enumerated at indicated time points. (D–F) CFU counts per gram of spleen derived from mice after subcutaneous infection with S119 Δ*asn*A (D), S119 *gln*P⁻ (E), and S119 *asn*B⁻ (F) compared to S119 were determined at indicated time points. (G–I) Lesion areas of mice infected with S119 Δ*asn*A (G), S119 *gln*P⁻ (H), and S119 *asn*B⁻ (I) compared to S119 were determined at different time points post-infection. Data information: Five mice per group per data point were used (A–I). The values shown represent the means ± SD. Statistical analysis was performed using the Mann–Whitney *U* test (A–F) and two-way ANOVA (G–I). Source data are available online for this figure.

(Fig. EV5D–F). Mice challenged with the indicated mutants recovered more rapidly than mice challenged with the WT strain S119 (Fig. 5A–I) and (Fig. EV5A–F). The clearance rates of bacteria from the soft tissue and spleen and the decrease in lesion size and weight of the spleens of mice challenged with the mutants were more rapid and lighter than those of the WT S119 strain, respectively (Figs. 5A–I and EV5A–F). The in vivo data set concurs with the RT-qPCR determinations conducted for the indicated strains in CDM with or without Asn supplementation (Figs. 2 and 3) and the functional determinations of ScpA and ScpC activities (Fig. 4). Furthermore, an *asn*A⁻ mutant of GAS strain 854 was also

attenuated in the same model (Fig. EV5G), demonstrating that Asn-mediated regulation of M1T1 virulence is ubiquitous among GAS M1T1 strains.

## Metabolism of Asn controls intracellular ADP/ATP levels

To address the impact of Asn on the GAS metabolome, we assessed the level of intracellular and extracellular metabolites (Mackay et al, 2015). These experiments were conducted on the WT strain S119 and its derived *gln*P⁻ and *asn*B⁻ mutants, grown to OD$_{600}$ = 0.35 and 0.7 in CDM supplemented or not with Asn. The score plots of

the probabilistic principal component analyses (PPCA) conducted on intracellular metabolites of all the samples at both optical densities for the indicated pairs of strains demonstrate that groups (clusters) were formed, thus statistically validating these measurements (Fig. EV6A,B). The relative quantity of intracellular Asn in the S119 grown in the presence of Asn was significantly higher at $OD_{600} = 0.35$ than that of S119 grown in the absence of Asn. The same was not valid for $glnP^-$ that cannot take up Asn (Fig. 6A). The relative amount of Asn in $AsnB^-$ grown in CDM without Asn was almost as high as that of the S119 grown with Asn (Fig. 6A).

Moreover, the relative Asn level in the $asnB^-$ mutant grown in the presence of Asn was more than 80-fold higher than that of the WT grown under similar conditions, thus corroborating that the asparaginase activity of AsnB is very efficient (Fig. 6A). At $OD_{600} = 0.7$, the relative Asn levels of S119 and $glnP^-$ mutant grown in the absence and presence of Asn were comparable, whereas those of the $asnB^-$ mutant were significantly higher under both conditions (Fig. 6A). Extracellular Asn level showed that it was taken up entirely by the WT strain before the culture reached $OD_{600} = 0.35$ and remained negligible in the medium to $OD_{600} = 0.7$.

Furthermore, no Asn was detected in the extracellular medium when GAS was grown in CDM without Asn, suggesting that GAS does not release Asn to the medium during growth (Fig. EV6C). The $glnP^-$ mutant did not import Asn from the medium into the bacteria; thus, the relative extracellular Asn level remained constant and high (Fig. EV6C). The $asnB^-$ mutant imported about 30% of the extracellular Asn at $OD_{600} = 0.35$; almost all of it was absorbed when the culture reached $OD_{600} = 0.7$ (Fig. EV6C). The reduced rate of Asn uptake in the $asnB^-$ mutant compared to that of S119 probably results from a feedback inhibition exerted by the increased level of intracellular Asn (Fig. EV6C).

To follow the metabolic status of S119 grown in the presence and absence of Asn at $OD_{600} = 0.35$ and 0.7, we measured the relative levels of some of the intermediate metabolites along the metabolic pathways of GAS, leading to the formation of lactic acid from glucose (Pancholi and Caparon, 2022). We found that at $OD_{600} = 0.35$, the relationships between the levels of the denoted intermediates varied (Fig. 6B). However, at $OD_{600} = 0.7$, the relative levels of all metabolic intermediates were significantly higher for S119 grown in the presence of Asn, suggesting that Asn increased the rate of GAS metabolism (Fig. 6B). Indeed, the transcription of SP119_0416 encoding the glucose-6-phosphate isomerase was highly upregulated in the presence of Asn (Fig. EV1B). Furthermore, the transcriptions of the genes encoding the ATP synthase (SP119_0124, SP119_0125, SP119_0126, SP119_0127, SP119_0128 SP119_0129) were also significantly upregulated in the presence of Asn (GEO accession number GSE268517). Thus, it appears that Asn upregulates glucose metabolism to gain higher energy, which is committed to enhanced growth and increased gene expression.

The glucose levels in the extracellular CDM were similar for S119 and derived mutants grown in CDM supplemented or not with Asn to $OD_{600} = 0.35$ and 0.7 (Fig. EV6D). Furthermore, the relative intracellular glucose levels at $OD_{600} = 0.35$ and 0.7 were comparable for the indicated strains, suggesting that glucose availability is not a limiting factor for GAS metabolism and growth (Fig. EV6E).

Nevertheless, Asn strongly affects ATP levels. In the presence of Asn, the level of ATP was the lowest for S119 compared to its $glnP^-$ and $asnB^-$ derived mutants, both at $OD_{600} = 0.35$ and $OD_{600} = 0.7$

(Fig. 6C). It was reported that when ADP levels exceed those of ATP, it stimulates the phosphatase activity of structurally related CovS sensors, (Castelli et al, 2000; Igo et al, 1989; Sanowar and Le Moual, 2005; Zhu et al, 2000). Therefore, we determined the ADP/ATP ratio and found that in the presence of Asn at both $OD_{600} = 0.35$ and 0.7, the ADP level of S119 grown in CDM supplemented with Asn is significantly higher than that grown in CDM only (Fig. 6D). However, mutants in GlnPQ and AsnB, grown in CDM supplemented with Asn possess lower ADP/ATP ratio like that of S119 grown in the absence of Asn (Fig. 6D).

## Discussion

GAS is a highly adapted and human-restricted pathogen. It is also multiple-amino-acid-auxotrophic (Davies et al, 1965). Therefore, it must derive some of its nutritional resources from the human host during infection. Nonetheless, although GAS synthesizes Asn via AsnA, it also scavenges Asn from the host, suggesting that its AsnA activity is sluggish and does provide the demand of GAS for Asn during the infection. Whereas the mechanism of Asn acquisition from the host has been extensively studied (Anand et al, 2021; Baruch et al, 2014), we know relatively little about Asn metabolism in GAS and how it is linked to virulence. Usually, host-adapted pathogenic bacteria produce virulent factors capable of causing reversible minor damage to host cells at low doses, thus facilitating the attainment of essential nutrients. However, when produced in excess, these factors may cause irreversible damage, leading to death of both the host and the bacterial during invasive infections such as NF and STSS (Brouwer et al, 2023; Cole et al, 2011). Therefore, nutrient availability is expected to tightly. control virulence factor production.

Here, we show that the presence of Asn in a minimal chemically defined medium increases virulence gene expression in a CovR-dependent fashion. It also decreases the transcription of AsnA, the ABC transporter exclusively responsible for Asn uptake GlnPQ, and that of the hemolysin toxins responsible for scavenging Asn from the host. In addition, we demonstrate that Asn metabolism is a fundamental process that controls GAS virulence. The uptake of Asn by GlnP and its synthesis by AsnA are ATP-consuming reactions that reduce the intracellular level of ATP, thus increasing the ADP/ATP ratio since GAS exclusively utilizes glycolysis (Pancholi and Caparon, 2022), providing only two molecules of ATP formed for each consumed glucose molecule.

ADP stimulates the phosphatase activity in structurally related TCS to CovS (Castelli et al, 2000; Igo et al, 1989; Sanowar and Le Moual, 2005; Zhu and Inouye, 2002; Zhu et al, 2000). Thus, we postulate that the ratio of ADP/ATP also affects the level of CovR dephosphorylation/phosphorylation. Hence, in the presence of an excess of ADP, CovR dephosphorylation occurs, and it exerts a significant impact on the regulation of GAS genes, including many virulence factors (Fig. 6E). Because the mutants of AsnA, GlnPQ, and AsnB are attenuated in the murine model of human NF and exhibit increased ATP/ADP ratio in CDM compared to WT strain S119 we predict that the same mechanism exists in vivo.

We also demonstrate that Asn regulates the expression of SLO and SLS. In contrast to the many virulence factors that extracellular Asn upregulates due to dephosphorylated CovR, SLO and SLS are downregulated under similar conditions. This mode of regulation is probably achieved by the feedback inhibition of Asn, which also

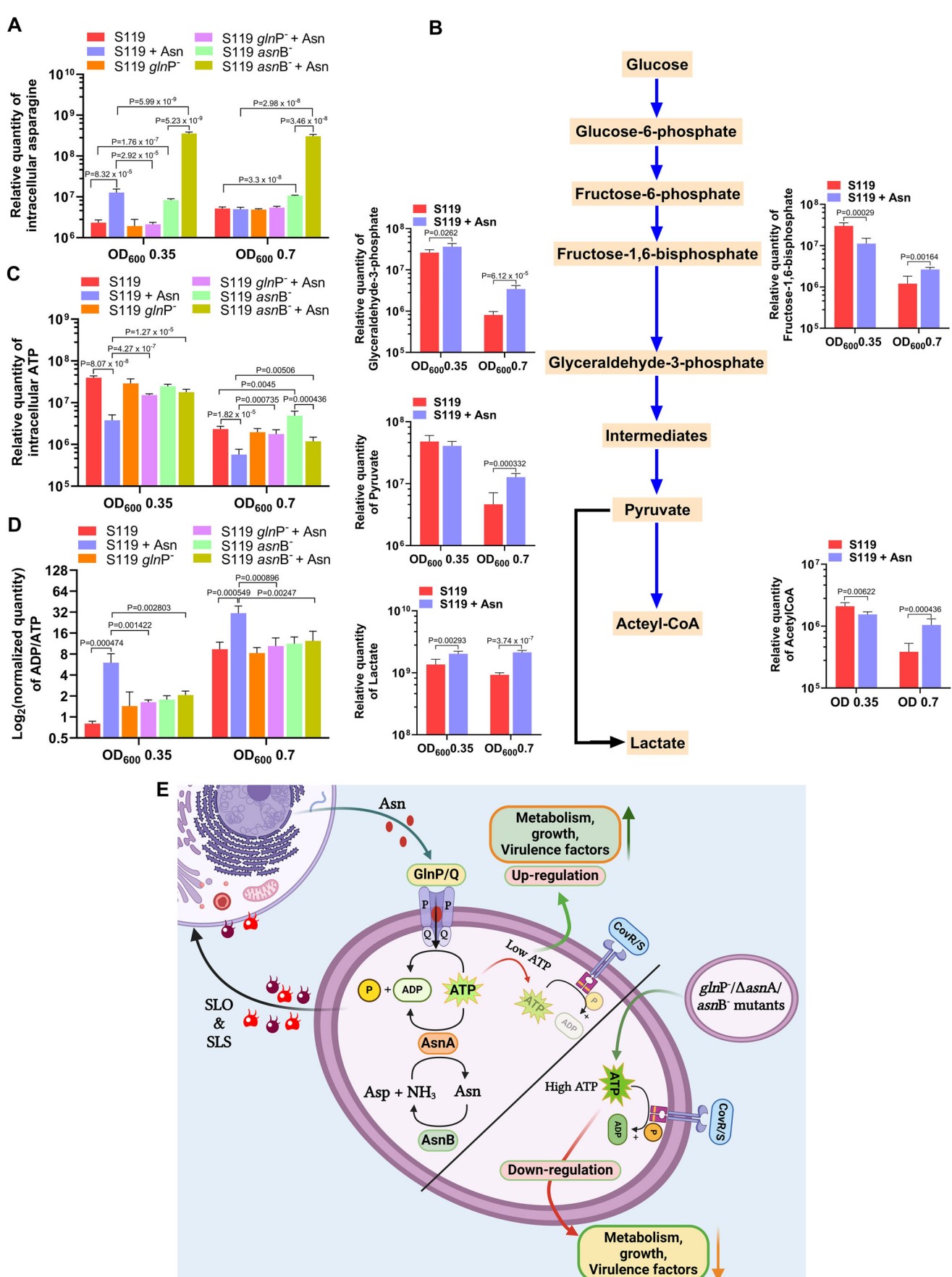

**Figure 6.    Elucidation of the mechanism by which Asn metabolism controls GAS virulence.**

(A, C) Intracellular metabolites: The relative amounts of intracellular Asn (A) and ATP (C) of the indicated strains grown in CDM or CDM supplemented with Asn were determined. (B) Determination of the intracellular intermediates along the glycolytic metabolic pathway: The measurements were conducted for the S119 strain grown in CDM or CDM-supplemented with Asn to $OD_{600} = 0.35$ or 0.7. (D) The indicated strains' intracellular ADP/ATP ratio grown in CDM or CDM supplemented with Asn was determined. (E) Schematic representation of the mechanism controlling the regulation of CovR phosphorylation by modulating the intracellular ADP/ATP ratio in response to Asn in the WT S119 and its derived mutants. Data information: Five biological replicates were used (A–D). The values shown represent the means ± SD. Statistical analysis was performed using an unpaired two-tailed $t$ test (A–D). Source data are available online for this figure.

downregulates the expression of GlnPQ and AsnA. These findings make perfect physiological sense because both toxins trigger ER stress and UPR, which upregulates the formation of Asn in infected host cells (Baruch et al, 2014). Furthermore, GlnPQ, which translocates Asn into GAS cells (Fig. 3B), is also regulated similarly. Thus, when Asn is in excess, extracellular and intracellular Asn formation decreases, and vice versa.

There are still several open questions to be answered. For example, what is the relationship between Asn-mediated gene regulation and GAS growth? However, we could genetically separate between the Asn-mediated gene regulation and growth as the ΔasnA mutant grew well in the presence of Asn but could not regulate virulence genes and lacked the negative AsnA-feedback regulation. Nonetheless, Asn-mediated growth enhancement may involve regulatory circuits that are partially dependent on CovR phosphorylation or AsnA-feedback regulation. We previously reported that genes linked to replication, such as polA, lig, danX, and others, were upregulated by Asn (Baruch et al, 2014). The comparison of RNA-seq data of covR, covS, asnA, glnPQ, and asnB mutants should address the above question more directly.

Another unresolved issue raised by our study is the identity of the kinase responsible for CovS-independent phosphorylation of CovR, which is almost eliminated in CDM supplemented with Asn. It was suggested that acetyl phosphate plays a role in CovR phosphorylation in vivo for *Streptococcus mutans*, which natively does not possess CovS (Khara et al, 2018). The ATP levels in S119 grown in CDM in the presence of Asn are low and limiting (Fig. 6B). The formed acetyl phosphate concentration should be low under these conditions and not reach the mM concentration required for phosphorylation. Thus, possibly non-cognate sensor kinases or an orphan histidine-kinase regulator cause the CovR phosphorylation through a crosstalk mechanism, and they respond to changes in the intracellular ADP/ATP concentrations similarly.

Several hematopoietic cancer cells cannot synthesize Asn or synthesize it sluggishly, which is insufficient to meet their metabolic requirements. Therefore, nutritional inhibition targeting Asn is used as an anti-cancer strategy, and Gram-negative bacterial asparaginases, like Kidrolase, have been applied to treat acute lymphoblastic leukemia (ALL) (Tabe et al, 2019; Yuan et al, 2024). We used Kidrolase successfully to treat intraperitoneal GAS infection in a mouse model, and Kidrolase prevented GAS growth in human blood (Baruch et al, 2014).

Another metabolic feature cancer cells share with GAS is the shift toward glycolysis and lactate production (the Warburg effect). This metabolic change occurs even in the presence of oxygen and fully functioning mitochondria. The primary role of residual respiration is to provide aspartate, which serves as a substrate for nucleotide synthesis (Sullivan et al, 2015). When drugs such as metformin inhibit the electron transport chain, it also limits the Asn level and impairs mTOR complex I (mTORC1) activity, thus reducing cell proliferation. An exogenously added Asn restores mTORC1 activity and, hence, cell proliferation. These findings suggest that compounds limiting Asn formation, uptake, and accelerating Asn degradation can be harnessed for therapeutic benefit together with metformin (Krall et al, 2021).

In this paper, we suggest that compounds limiting Asn formation, uptake, and accelerating Asn degradation can also be harnessed to treat GAS-invasive diseases because they would decrease the ratio of ADP/ATP, thus increasing CovR phosphorylation and decreasing virulence. The mainstay of treatment of invasive GAS infections is surgical debridement of infected tissues, prompt administration of intravenous antibiotics, and supportive care (Allaw et al, 2024; Allen and Moore, 2010; Anaya and Dellinger, 2007). Despite this, the associated mortality for invasive GAS diseases remains high, ranging from 23% to 35% in resource-rich settings (Cole et al, 2011; Walker et al, 2014). Furthermore, no safe and universally available vaccine against GAS exists (Dale and Walker, 2020). Therefore, the necessity to develop effective novel treatments against GAS infections is self-evident.

## Methods

**Reagents and tools table**

| Strain | Description | Source |
|---|---|---|
| **GAS strains** | | |
| S119 | An invasive strain isolated from human blood in 2008 | |
| S119 ΔcovS | | This study |
| S119 covR | | This study |
| S119 ΔasnA | | This study |
| S119 ΔrocA | | This study |
| S119 ΔasnA - pLZ asnA | | This study |

| Strain | Description | Source |
|---|---|---|
| S119 Δ*asn*A *R101K | | This study |
| S119 Δ*asn*A - p*LZ asn*A *R100K | | This study |
| S119 *gln*P⁻ | | This study |
| S119 *gln*P⁻ - p*LZ gln*P | | This study |
| S119 *asn*B⁻ | | This study |
| S119 *asn*B⁻ - p*LZ asn*B | | This study |
| GAS 854 | M1T1 GAS clinical isolate from a patient with a retroperitoneal abscess | (4) |
| GAS 854 *asn*A⁻ | | This study |
| GAS 5448 | M1T1 GAS clinical isolate from a patient with necrotizing fasciitis and toxic shock syndrome | (5) |
| **Primer name** | **Primer's sequence (5'-3')** | **Source** |
| **RT-RT-PCR** | | |
| *gyr*A-RT-F | TTATCACGTTCCAAACCAGTC | (6) |
| *gyr*A-RT-R | CGACTTGTCTGAACGCCAAA | (6) |
| *mga*-RT-F | TGCGTTTGATAGCATCAAACAAG | This study |
| *mga*-RT-R | CAAGGAGATGAACCCAGTTGGT | This study |
| *emm*-RT-F | GCAGAAGCAAAAGCACTCAAAG | This study |
| *emm*-RT-R | TTTCCAGCTCTTAGTTTTGCAAGTT | This study |
| *scp*A-RT-F | TGCAGCGGCAGACTCAAC | This study |
| *scp*A-RT-R | TTTGACCGTAGCAGTTTCAGTGA | This study |
| *scp*C-RT-F | TAATACAGGTCCTGATGCGACTCA | This study |
| *scp*C-RT-R | CAATCGGTTTACCATCTGCATC | This study |
| *ska*-RT-F | GCGAACGTAACTTAGACTTCAG | This study |
| *ska*-RT-R | GATACGGTTGGTGTCATC | This study |
| *has*A-RT-F | ACCGTTCCCTTGTCAATAAAGG | This study |
| *has*A-RT-R | AACGTCAGCGTCAGATCTTTCA | This study |
| *nga*-RT-F | ACCGCATCACATTGAGATTGAT | This study |
| *nga*-RT-R | TCCCTGATGGACCTCTGTTACC | This study |
| *hyl*A-RT-F | CAGGCATTTGCTTGGCATAA | This study |
| *hyl*A-RT-R | CCAGGAGAGCTTCTTCCACTTC | This study |
| *slo*-RT-F | AAAACAAACCAGACGCGGTAGT | This study |
| *slo*-RT-R | TTGTCTCCCATACCTGGTAAATCA | This study |
| *sag*A-RT-F | TACCACCTTGAGAATTACCA | This study |
| *sag*A-RT-R | AGGAGGTAAACCTTATGTTA | This study |
| *gln*P-RT-F | ATG TCA AGG CAT TTG ATG ATG G | This study |
| *gln*P-RT-R | CGA CGA CCT TGA GAG ATA GC | This study |
| *asn*B-RT-F | ATG CAT TTC TGG AAC CAT CCC | This study |
| *asn*B-RT-R | CCC TAG AGC TTC GAT AAC TAG ACC | This study |
| *asn*A-RT-F | CTCAATGGGAATCCGTGT | This study |
| *asn*A-RT-R | CAAGAACATGGCCATACGTG | This study |
| *sra*-f | CTGATGCTACTGCCATAGCAG | (7) |
| *sra*-r | GCGTTCAGGAAGTCTAGCTC | (7) |

| Strain | Description | Source |
|---|---|---|
| **Primers used for mutant generation** | | |
| **Primer name** | **Primer sequence (5'-3')** | **Remarks** |
| *asn*B start-*Hind*III-L | CGTAAGCTTCTGGTGGAACCATTTCTATG | Amplifies 472 bp of *asn*B starting at nucleotide 25 after the start codon. |
| *asn*Bstart-*Pst* | CGTCTGCAGGTTGAAATATTGGTTGTATG | |
| *asn*B comp-F *Eco*RI | GTCTGAATTCTCTTTTGCCTCCTGTGTTTATGATAG | Amplifies a 3145 bp fragment containing *asn*B and its upstream region |
| *asn*B comp-R-*Bam*HI | TCAGGATCCGTGGGACATTTCTAAAGGAAGCTAGA | |
| *gln*P-inact-F | AAGCGAGTGCTCTAATGGC | Amplifies a 671 bp fragment starting at nucleotide 249 of *gln*P |
| *gln*P-inact-R | ACCTTGTTGTTTAGCAATAGCTTTGATTAA | |
| *gln*PQcomp-F-*Bam*HI | TCAGGATCCACTATTTTAACATAGATAAGCCAGAATCACCT | Amplifies a 3145 bp fragment containing *gln*PQ and its upstream region |
| *gln*PQcomp-R-*Eco*RI | GACAGAATTCCAAGGAAACCTGTTTCCTTGCC | |
| *cov*R-F2 | CGCGGATCCCATGATGATGACTGCGCGTGA | Amplifies a 296 bp fragment starting at nucleotide 227 of *cov*R |
| *cov*RS-p3-F | GATGTCTATATTCGTTATCTCC | |
| *cov*R-R3 | CGCAAGCTTCATGACACGATTCATATTAGTC | |
| *cov*RS-p9-R | CTTGTGCCAAATAACTCAACA | |
| Gibson-p*LZ* Km-F | CGACATATCGGATTGTCCCTATACG | |
| Gibson-p*LZ* Km-R | CGTATAGGGACAATCCGATATGTCG | |
| Gibson-p*LZ* *asn*A-Arg100-Lys-F | GGTCAAGTGAATCCTCATCAGGTTTGAGAGCCTTCATATTAACAACAAGG | |
| Gibson-p*LZ* *asn*A-Arg100-Lys-R | CCTGATGAGGATTCACTTGACC | |
| *roc*A up-F | GCCATTCCTATCTCCGCAGATAAGCTC | Amplifies a 3298 bp fragment containing *roc*A and its upstream and downstream regions |
| *roc*A down-R | GCTTTGACAACGCTTACTATGAGG | |
| *asn*Aup-L-*Xba*I | CGTCTAGACGCAGGGTAATCAAGAGG | Amplifies a 411 bp fragment upstream of *asn*A |
| *asn*Aup-R-*Sma*I | CCCGGGACAACATCTAACTTAGCAA | |
| *asn*ADown-R-*Pst*I | CTCAACCAAAACTGCAGCAGACATC | Amplifies a 380 bp fragment downstream of *asn*A |
| *asn*ADown-L-*Sma*I | GCCCGGGTATGGCCTCAAGAGGTTCG | |
| *asn*ADown -R | CTCAACCAAAACTGCAGCAGACATC | Amplifies with primer *asn*Aup-L-*Xba*I, a 1703 bp fragment containing *asn*A and its upstream and downstream regions |
| **Reagent/resource** | **Reference/source** | **Identifier/catalogue number** |
| **Antibodies** | | |
| StarBright Blue 700 Goat Anti-Mouse IgG | Bio-Rad Laboratories | Cat#12004159 |
| Rabbit anti-CovR antibody | Provided by Samuel A. Shelburne III, Department of Infectious Diseases, MD Anderson Cancer Center, Houston, Texas, USA | NA |
| **Chemicals** | | |
| L-Asparagine | Sigma-Aldrich | Cat# A0884 |
| β-Nicotinamide adenine dinucleotide sodium salt | Sigma-Aldrich | Cat# N0632 |
| Para-aminobenzoic acid | Sigma-Aldrich | Cat# A9878 |
| Biotin | Sigma-Aldrich | Cat# B4501 |
| Folic acid | Sigma-Aldrich | Cat# F7876 |
| Niacinamide | Sigma-Aldrich | Cat# N5535 |
| Calcium pantothenate | Sigma-Aldrich | Cat# C8731 |
| Pyridoxal hydrochloride | Sigma-Aldrich | Cat# P9130 |
| Pyridoxamine dihydrochloride | Sigma-Aldrich | Cat# P9380 |

| Strain | Description | Source |
|---|---|---|
| Riboflavin | Sigma-Aldrich | Cat# R4500 |
| Thiamine hydrochloride | Sigma-Aldrich | Cat# 47858 |
| Vitamin B12 | Sigma-Aldrich | Cat# V2876 |
| Adenine | Sigma-Aldrich | Cat# A8626 |
| Guanine hydrochloride | Sigma-Aldrich | Cat# 51030 |
| Uracil | Sigma-Aldrich | Cat# U0750 |
| Donkey serum | Sigma-Aldrich | Cat# D9673 |
| Normal mouse serum | Sigma-Aldrich | Cat# M5905 |
| 4',6-diamidino-2-phenylindole (DAPI) | Thermo Scientific | Cat# R37606 |
| Octylphenoxypolyethoxyethanol (Nonidet P-40) | Sigma-Aldrich | Cat# N-6507 |
| Complete Mini, EDTA-free protease inhibitor cocktail tablets | Roche Molecular Diagnostic, USA | Cat# 11836170001 |
| PhosSTOP™ | Roche Molecular Diagnostic, USA | Cat# 04906845001 |
| **Critical commercial assays** | | |
| SV Total RNA isolation system | Promega corporation | Cat# Z3100 |
| RevertAid First Strand cDNA Synthesis Kit | Thermo Scientific | Cat# K1621 |
| Direct-zol™ RNA miniprep kit | Zymo-Research | Cat# R2050 |
| RQ1 RNase-Free DNase | Promega corporation | Cat# M6101 |
| M-MLV Reverse Transcriptase | Promega corporation | Cat# M5313 |
| 2x Tamix Fast SyGreen Mix Hi-ROX | Tamar Laboratory Supplies Ltd. | Cat# TA20.12-05 |
| Pan-Bacteria (RNA-Seq) riboPoolTM kit | siTOOLs BioTech, Germany | Cat# dp-K012-00026 |
| KAPA Stranded mRNA-Seq Kit | Kapa Biosystems, Wilmington, USA | Cat# KK8421 |
| **Experimental models: organisms/strains** | | |
| Mouse: BALB/c OlaHsd 3-4 weeks old female 10–12 grams | Envigo RMS Ltd. (Israel) | N/A |
| **Software and algorithms** | | |
| GraphPad Prism 5 | GraphPad | https://www.graphpad.com/scientificsoftware/prism/ |
| Vector NTI Suite | InfoMax | https://www.thermofisher.com/il/en/home/life-science/cloning/vector-ntisoftware.html |
| ImageJ software | National Institutes of Health and the Laboratory for Optical and Computational Instrumentation | https://imagej.nih.gov/ij/ |
| FastQC v0.11.9 | N/A | http://www.bioinformatics.babraham.ac.uk/projects/fastqc/ |
| Cutadapt v2.10 | N/A | http://cutadapt.readthedocs.org/en/stable/ |
| fastq_quality_filter v0.0.14 | N/A | http://hannonlab.cshl.edu/fastx_toolkit/ |
| bowtie2 version 2.3.4.3 | N/A | https://bowtie-bio.sourceforge.net/bowtie2/index.shtml |
| htseq-count v0.6.0 | N/A | http://www-huber.embl.de/users/anders/HTSeq/doc/count.html |
| DESeq2 package v1.26.0 | N/A | https://bioconductor.org/packages/release/bioc/html/DESeq2.html |

| Strain | Description | Source |
|---|---|---|
| R version 3.6.1 | with packages RColorBrewer_1.1-2, pheatmap_1.0.12, ggplot2_3.2.0 and ggrepel_0.8.1 | https://www.npackd.org/p/r/3.6.1 |
| **Other** | | |
| Mic qPCR Cycler | Bio-Molecular Systems | N/A |
| NanoDrop™ One | Thermo Scientific | N/A |
| Trans-Blot® Turbo™ Transfer System | Bio-Rad Laboratories | N/A |
| Zeiss AxioImager Z1 fluorescence microscope | Zeiss | N/A |
| Zeiss AxioCam HRm Rev 3 Digital Camera | Zeiss | N/A |
| Polytron™ PT 2100 homogenizer | Kinematica AG | N/A |
| 8.0-mm punch biopsy | Acuderm Inc. | Cat# P850 |
| Digital caliper | Bar Naor Ltd. | Cat# BN30087-00 |
| Poly-lysine slides | Thermo Scientific | Cat# P4981 |
| Millex-GV Syringe Filter Unit, 0.22 µm, PVDF, 33 mm. | Merck Millipore Ltd. | Cat# SLGV033RS |
| Bacto™ Todd Hewitt Broth | Becton, Dickenson, and Company | Cat# 249240 |
| Bacto™ Yeast Extract | Becton, Dickenson, and Company | Cat# 212750 |
| Blood agar plates | hylabs® | Cat# PD049 |
| RNA protect bacteria reagent | Qiagen | Cat# 1018380 |
| 4–15% Mini-PROTEAN® TGX™ Precast Protein Gels | Bio-Rad Laboratories | Cat# 4561084 |
| 4x Laemmli Sample Buffer | Bio-Rad Laboratories | Cat# 1610747 |
| Trans-Blot Turbo Transfer Pack | Bio-Rad Laboratories | Cat# 1704157 |
| Mutanolysin | Sigma-Aldrich | Cat# 55466-22-3 |
| Lysozyme | Fischer Scientific | Cat# PI89833 |
| SYBR green | Thermo Scientific | Cat# AB4162 |
| Fetal Bovine Serum (FBS) | Biological Industries | Cat# 04-121-1A |
| Human IL-8 Quantikine® ELISA kit | R & D Systems | Cat# D8000C |
| 16.5% Mini-PROTEAN® Tris-Tricine Gel | Bio-Rad Laboratories | Cat# 4563063 |
| Human Interleukin-8 (rhIL-8/CXCL-8) Recombinant Protein | R & D Systems | Cat# 208-IL |
| Low-range Rainbow molecular weight marker | DEUTSCHER, France | Cat# RPN755E |
| BLUeye Prestained Protein Ladder, | Sigma-Aldrich | Cat# 94964 |
| Instant Blue | Expedeon Inc. | |
| Tris Tricine SDS Running Buffer | Bio-Rad Laboratories | Cat# 1610744 |
| purified recombinant human complement component C5a | R & D Systems | Cat# 2037-C5-025/CF |
| Tris-Glycine Running Buffer | Bio-Rad, USA | Cat# 161-0772 |
| ß-mercaptoethanol | Sigma-Aldrich | Cat# M6250 |
| SuperSep™ Phos-tag™ Gels (Zn²⁺ and 12.5% with 50 µM of phostags, 17 wells) | Wako, Japan | Cat# 195-17991 |

## Bacterial culture

The GAS strains used in this study are represented in Reagents and Tools Table. Methodology and primers used for constructing all mutants' have been described in Reagents and Tools Table. GAS was cultured overnight without shaking in Todd-Hewitt broth supplemented with 0.2% yeast extract (THY) in sealed tubes at 37 °C. When necessary, antibiotics were added at the final concentrations of 250 µg/ml for kanamycin (Km), 50 µg/ml for spectinomycin (Spec), or 1 µg/ml for erythromycin (Em). The following morning, overnight cultures were diluted 1:20 and grown in THY medium with appropriate antibiotic, when needed, to an early-log phase (OD600 of 0.3), washed, and resuspended in chemically defined medium (CDM) designed by van de Rijn and Kessler (van de Rijn and Kessler, 1980). The growth rates of different bacterial strains were determined in CDM supplemented with or without asparagine (Asn) at different concentrations (2–100 µg/ml). 1 ml of freshly prepared CDM with bacteria (in the absence or presence of Asn and an appropriate antibiotic) was added to each well of a 24-well plate, and the plate was incubated at 37 °C in a 5% $CO_2$ atmosphere. The absorbance was measured at $OD_{600}$ at regular time intervals. This method of culturing GAS in CDM was followed for all experiments.

## Construction of GAS mutants

All mutants used in this study were generated using different primers (Reagents and Tools Table). **S119 ΔcovS**—a 718 bp fragment containing 107 bp of the end and 611 bp of the beginning of covS was amplified with the primers covRS-p3-F and covR-R3 using S119 genomic DNA as a template. The PCR product was cloned into pGEM-T easy vector and then digested with EcoRV, and a Km-resistance gene (ΩKm) was cloned on this site. The resulting insert containing covS upstream and downstream sequences separated by ΩKm gene was released from pGEM-T easy with EcoRI, treated with DNA Polymerase I, Large (Klenow) Fragment (NEB) for blunting the fragment, and then cloned into pJRS233 digested with EcoRI. The resulting plasmid, pJRScovS-ΩKm, was electroporated into strain S119 for knockout of covS. **S119 covR⁻**—a 296 bp fragment of covR was amplified with the primers covR-F2 and covR-R3 using S119 gDNA as a template. The PCR product was cloned into pJRS233, which was digested with HindIII and BamHI. The resulting plasmid, pJRScovR, was electroporated into strain S119 for insertional inactivation of covR. **S119 asnB⁻**—a 472 bp fragment of asnB was amplified with the primers asnB start-HindIII-L and asnBstart-Pst using S119 gDNA as a template. The PCR product was cloned into pJRS233, and the resulting plasmid pJRSasnBstart was electroporated into strain S119 for insertional inactivation of asnB. **S119 asnB⁻–pLZ asnB**—a 1071 bp fragment including the gene asnB and its upstream region was amplified with the primers asnB comp-F EcoRI and asnB comp-R-BamHI using S119 gDNA as template. The PCR product was cloned into pLZ12, and the resulting plasmid pLZasnB comp was electroporated into strain S119 asnB⁻ strain to complement the asnB⁻ mutant. **S119 glnP⁻**—a 671 bp fragment of glnP was amplified with the primers glnP-inact-F and glnP-inact-R using S119 gDNA as a template. The PCR product was cloned into pJRS233, and the resulting plasmid pJRSglnPinact was electroporated into strain S119 for insertional inactivation of glnP. **S119**

**glnP⁻–pLZ glnPQ**—A 3152 bp fragment glnPQ and its upstream region were amplified with the primers glnPQcomp-F-BamHI and glnPQcomp-R-EcoRI using S119 gDNA as template. The PCR product was cloned into pLZ12, and the resulting plasmid pLZglnPQ comp was electroporated into strain S119 glnP⁻ to complement glnPQ. **S119 ΔasnA**—a fragment containing 411 bp upstream of asnA, 380 bp downstream of asnA separated by ΩKm gene, was cloned into pJRS233, and the resulting plasmid pJRSasnA was electroporated into strain S119 for knockout of asnA. The 411 bp upstream fragment was amplified using the primers asnAup-L-XbaI and asnAup-R-SmaI and GAS JS95 gDNA as a template. The 380 bp downstream fragment was amplified using the primers asnADown-R-PstI and asnADown-L-SmaI using JS95 gDNA as a template. **S119 ΔasnA- pLZ asnA**—a 1703 bp fragment including the gene asnA and its upstream and downstream regions was amplified using the primers asnAup-L-XbaI and asnADown–R using S119 genomic DNA as a template. The PCR product was cloned into pLZ12, and the resulting plasmid pLZasnA comp was electroporated into strain S119 ΔasnA for complementation of asnA. **S119 ΔasnA- pLZ asnA R100K**—pLZ asnA R100K was constructed by amplifying pLZasnA with 2 sets of primers: Gibson-pLZ km-F + Gibson-pLZ asnA-Arg100-Lys-R and Gibson-pLZ Km-R + Gibson-pLZ asnA-Arg100-Lys-F, using pLZasnA as a template. The amplified PCR products were ligated using a Gibson assembly kit (NEB, USA). The resulting plasmid pLZ asnA R100K was electroporated into strain S119 ΔasnA for expression of mutated asnA. **S119 ΔrocA**—a 3298 bp fragment containing rocA flanked by its upstream and downstream sequences was PCR amplified with the primers rocA up-F and rocA down-R using S119 gDNA as a template and was cloned into pGEM®-T Easy Vector. The resulting plasmid pGEM rocA was digested with XbaI and HpaI, releasing a 971 bp fragment of rocA, and a km-resistance gene was inserted. The resulting fragment containing rocA upstream and downstream sequences separated by ΩKm gene was cloned into pJRS233, and the resulting plasmid pJRSRrocA was electroporated into strain S119 for knocking out rocA.

## Extraction of RNA from GAS and qRT-PCR determinations

The total RNA of GAS was isolated using the phenol-ethanol extraction method, and purification was conducted using the Direct-zol RNA miniprep kit (Zymo Research). The RNA concentration and purity were evaluated using NanoDrop One (Thermo Scientific). RNA was treated with RQ1 DNase (Promega) according to the manufacturer's instructions to avoid genomic DNA contamination. According to the manufacturer's instructions, m-MLV Reverse Transcriptase (Promega) was used for reverse transcription. For real-time PCR, cDNA was diluted, and quantitative PCR was performed on Mic qPCR Cycler (Bio-Molecular Systems) using the 2× Tamix Fast SyGreen Mix Hi-ROX (Tamar Laboratory Supplies Ltd). The primers used are listed in the Reagents and Tools Table. Each target gene's expression amounts were normalized to gyrA and analyzed using the 2-$\Delta\Delta C_T$ method (Livak and Schmittgen, 2001).

## RNA-Seq and data analysis

Total RNA was extracted from GAS using the phenol-ethanol extraction method, and purification was conducted using the Direct-zol RNA miniprep kit (Zymo Research). Ribosomal RNA

removal was achieved using the Pan-Bacteria (RNA-Seq) riboPool™ kit (siTOOLs BioTech, Germany). Sample quality was assessed using a 2100 Bioanalyzer (Agilent), and sample quantity was determined using a NanoDrop 8000 spectrophotometer (Thermo Scientific). According to the manufacturer's recommendations, the RNA-seq directional libraries were generated using the KAPA Stranded mRNA-Seq Kit (Kapa Biosystems, KK8421). A 76 bp single-read DNA sequencing was performed using the Illumina Nextseq500 platform (Core Research Facility, Faculty of Medicine, The Hebrew University of Jerusalem, Israel). Data were generated in the standard Sanger FastQ format, and raw reads have been deposited under BioProject GSE268517 with the Sequence Read Archive (SRA) at the National Center for Biotechnology Institute. The NextSeq basecall files were converted to fastq files using the bcl2fastq program with default parameters (without trimming or filtering applied at this stage). After ensuring quality, the processed reads were aligned to the reference genome of *Streptococcus pyogenes* strain S119 (GCA_900608505.1, Genebank).

## ScpC and ScpA cleavage assay

GAS was cultured in CDM in the absence or presence of Asn (10 μg/ml) or 300 nM LL-37 in 24-well plates at 37 °C in 5% $CO_2$ atmosphere, and samples were collected at late log phase ($OD_{600}$ of 0.7). The cell-free supernatants were incubated with an equal volume of 1 mg/ml of purified recombinant human IL-8 (R&D Systems, USA) and human complement component C5a (R&D Systems, USA) at 37 °C for 2 h. The samples were heated at 100 °C for 5 min with 4x Tricine loading buffer (0.4 M Tris HCl pH 6.8, 80% glycerol, 4% SDS, and 0.08% Coomassie blue) to stop the reaction. The proteins were resolved on precast 16.5% Mini-PROTEAN® Tris-Tricine gels using the Mini-PROTEAN gel apparatus (Bio-Rad). The samples were run on a low constant voltage at 4 °C for 6 to 8 h in Tris Tricine SDS Running Buffer (Bio-Rad, USA). Instant Blue (Expedeon Inc.) detected the peptides. The gels were de-stained with distilled water until the bands were visible.

## ELISA-based assessment of ScpC expression by performing IL-8 degradation assays

The samples from GAS culture were collected as mentioned above, and the cell-free supernatants were incubated with 1 ng/ml of recombinant human IL-8 (R&D Systems, USA) at 37 °C for 2 h. The samples were heated at 100 °C for 2 min; the reaction was stopped. IL-8 cleavage was represented as the residual amount of IL-8 in the samples, estimated using the Human IL-8 Quantikine ELISA kit (R&D Systems, USA).

## Protein extraction and Phos-tag western blotting

Bacteria were cultured in CDM without/with Asn (10 μg/ml) until the $OD_{600}$ reached 0.6–0.7. All further steps were performed at 4 °C. Bacterial cultures were collected, centrifuged, and pellets were washed with ice-cold sterile PBS. For protein extraction, the pellet was suspended in ice-cold lysis buffer [20 mM Tris-HCL pH 8 with 10 U mutanolysin (Sigma-Aldrich), cOmplete™, EDTA-free Protease Inhibitor Cocktail (Roche Molecular Diagnostic, USA), and PhosSTOP™ (Roche Molecular Diagnostic, USA) in PBS]. The bacterial cells were homogenized by using MagNA Lyser (Roche

Diagnostics). The lysates were centrifuged to collect the supernatants, and the total protein was quantified using Bradford's assay. Protein samples were mixed with 4x Laemmli Sample Buffer (900 μl of Laemmli Sample Buffer + 100 μl of ß-mercaptoethanol) and resolved in Phos-tag SuperSep Phos-tag Gels (Wako, Japan). Recombinant CovR was purified and phosphorylated in vitro as described and served as a control (Horstmann et al, 2011). The gel was treated three times with blotting buffer [10x SDS PAGE buffer 50 mL (5 of 1×) + MetOH 200 ml (2) + adjusted to a total volume of 1 liter (milliQ)] supplemented with 10 mM EDTA to remove all $Zn^{2+}$ from the gel. The blotting was performed using the Trans-Blot Turbo Transfer Pack (Bio-Rad). The membrane was blocked overnight at 4 °C in PBST with 5% skim milk. The membrane was washed and placed in a suspension of rabbit anti-CovR antibody in PBST (1:5000) for 1 h at room temperature. The membrane was washed and treated with fluorescently labeled Goat Anti-Rabbit IgG StarBright Blue 700 antibody (1:5000) in 10 ml of PBST buffer with 1% skim milk at room temperature for 2 h. Finally, the membrane was washed, and the fluorescently labeled proteins were visualized at a wavelength of 470 nm using the Biorad GelDoc system. The relative percentage of phosphorylated CovR was calculated using ImageJ software (open source, developed by NIH, USA).

## Asparaginase assay

The Nesslerization method was used to determine the intracellular L-asparaginase activity in whole-cell suspension (Farahat et al, 2020). GAS was grown in THY media until $OD_{600}$ reached 0.7. Cells were harvested, and the cell and the washed pellet were lysed with purified PlyC phage protein. An equal volume of lysate and reaction mixture (50 mM L-Asn and 100 mM Tris-HCl, pH 8.0) was mixed and incubated at 37 °C for 10 min. The reaction was terminated by adding 20 μL trichloroacetic acid (1.5 M). After centrifugation, 15 μL Nessler's reagent was added to the supernatant, and the absorbance was measured at 436 nm.

## Animals

In all, 3- to 4-week-old female BALB/c OlaHsd mice weighing 10–12 g were obtained from ENVIGO RMS (Israel Ltd.). Following the Hebrew University of Jerusalem's ethical guidelines, all procedures were performed for humane handling, caring for, and treating research animals (Protocol number MD-22-17143-5). Mice were kept in disposable cages supplemented with enrichment, and regular sterile food, water, and air were supplied separately in each cage. All cages were placed in specific pathogen-free (SPF) conditions during the experiment with controlled environmental conditions. The mice were left to acclimate for 3 days, after which treatment groups were randomized, and the littermates were evenly distributed in cages. Identification markings and shaving on dorsal flanks of already weighed mice were performed, and mice were infected. Following infection, twice daily, mice were given wet food and monitored based on parameters like body weight, activity level, fur, and eye appearance. As per the guidelines of the Institutional Animal Care Units of the Hebrew University's School of Medicine, based on the above parameters, a scoring method was implemented to decide humane endpoints where mice were euthanized according to ethically approved procedures.

## Sublethal murine model of human GAS soft tissue infection

Three- to 4-week-old female BALB/c OlaHsd mice weighing 10–12 g were obtained from ENVIGO RMS (Israel Ltd.). Following the Hebrew University of Jerusalem's ethical guidelines, all procedures were performed for humane handling, caring for, and treating research animals (Protocol number MD-22-17143-5). The murine model of human soft-tissue infection was injected with a sub-lethal dose ($5 \times 10^7$ CFU) of GAS strains injected subcutaneously (SC) into the rear flank of mice, and CFU counts were determined in soft skin and spleen samples. At various times, mice were euthanized by inhalation of isoflurane followed by cervical dislocation, and skin and spleen samples were collected. Skin tissue from the injection site was collected using a punch biopsy tool (Acu-Punch, Acuderm Inc.), and spleen samples were excised and transferred to 2 ml Eppendorf tubes containing 0.5 ml of sterile PBS. Tissues were homogenized, diluted, and plated on blood agar plates for all experiments, and CFUs were counted after overnight incubation at 37 °C. CFU counts were normalized to the weight of the soft tissue. The CFU counts for S119 $glnP^-$ and S119 $asnB^-$ mutants were matched with parallel plating on erythromycin (1 µg/µl) supplemented THY agar plates to check the stability of insertional inactivation. For determination of the lesion area, the dermonecrotic skin lesions were measured daily using a digital caliper (Bar Naor Ltd.). The lesion area was calculated with the formula A = (π/2)(length)(width) (Anand et al, 2021).

## Metabolomics

GAS (S119 and S119 $glnP^-$) were cultured in CDM without Asn until the OD600 reached 0.2 for time-dependent Asn uptake kinetics. Asn was added (final concentration of 10 µg/ml) in culture, and the samples were collected at different time points (5, 30, and 60 min). The sterile filtrate was mixed with a chilled extraction solution (50% methanol, 30% acetonitrile, 20% water) and stored at −80 °C until further analysis. For extracellular and intracellular metabolites detection, GAS (S119, S119 $glnP^-$, S119 $asnB^-$) were grown in CDM without or with Asn (10 µg/ml) until the $OD_{600}$ reached 0.35 or 0.7. For extracellular metabolite detection, samples were collected, and sterile filtrate was mixed with a chilled extraction solution until further analysis. Pellets obtained from 2 ml of cultures were washed and lysed for intracellular metabolite detection by pure PlyC phage protein. As shown above, the lysate was mixed with a chilled extraction solution (1:10). All samples were stored at −80 °C until further analysis. The Bradford assay was used to estimate protein in intracellular samples. The protein quantity in each sample was used to normalize the quantity of detected metabolites. All experiments were conducted in five biological replicates ($n = 5$). LC-MS metabolomics analysis was performed as described previously (Mackay et al, 2015) with slight changes for polar metabolite detection. In brief, the Vanquish ultra-high-performance liquid chromatography (UHPLC) system coupled to the Exploris 240 Orbitrap Mass Spectrometer (ThermoFisher Scientific) was used to resolve and separate the compounds. All metabolites were detected, and Xcalibur (ThermoFisher Scientific) was used to acquire data. Skyline version 23.1.1.503 generated chromatograms for each compound, and chromatographic peaks were inspected and

integrated. Relative quantification between sample groups was performed using the area of the signal. Metabolite AutoPlotter 2.6 (Pietzke and Vazquez, 2020) and Metaboanalyst were used for data visualization.

## Statistical analysis

GraphPad Prism version 10 software was used to plot results and perform statistical analysis. All values were represented as means ± standard deviation (SD). Data in bar graphs were analyzed using parametric unpaired two-tailed $t$ tests unless specified, and two-way ANOVA with Tukey post-tests and Mann–Whitney $U$ test, where indicated. In all figures, $P$ values were calculated to confirm the significance.

## Data availability

The reported RNA-Seq data are available in the NCBI Gene Expression Omnibus under accession GSE234272 (https://www.ncbi.nlm.nih.gov/geo/query/acc.cgi?acc=GSE268517).

The source data of this paper are collected in the following database record: biostudies:S-SCDT-10_1038-S44319-025-00447-z.

## Peer review information

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

## Acknowledgements

We thank the core facility of the Faculty of Medicine at Hebrew University for conducting the RNA-seq analysis. We are indebted to Dr. Abed Nasereddin (Core Research Facility, Faculty of Medicine, The Hebrew University of Jerusalem) and Dr. Nevo Yuval (Unit of Bioinformatics, Faculty of Medicine) for their valuable contributions. We also greatly appreciate the support of Bella Agranovich and Ifat Abramovich from the Laura and Isaac Israel, Perlmutter Metabolomics Center, a part of the Biomedical Core Facility at the Technion, Israel Institute of Technology, Haifa, for their assistance in metabolomics experimental design, sample processing, and data analysis. The authors thank Prof. Herve Bercovier from the Department of Microbiology and Molecular Genetics, The Institute for Medical Research, Israel-Canada (IMRIC), Faculty of Medicine, The Hebrew University of Jerusalem, for reading this manuscript and for his valuable suggestions. This work was supported by the Israeli Ministry of Innovation, Science and Technology grant (Grant number 0005663) and the Israel Science Foundation (ISF) grant (1926/24) to EH, NIH/NIAID grant R01-AI047928 to KSM and National Key R&D Program from the Ministry of Science and Technology of China (2023YFE0113500), Chinese National Natural Science Foundation (82202525) to XZ.

## Author contributions

**Abhinay Sharma**: Data curation; Formal analysis; Validation; Methodology; Writing—review and editing. **Aparna Anand**: Data curation; Formal analysis; Validation; Methodology; Writing—review and editing. **Miriam Ravins**: Data curation; Methodology; Project administration. **Xiaolan Zhang**: Funding acquisition; Writing—review and editing. **Nicola Horstmann**: Resources; Data curation. **Samuel A Shelburne**: Resources; Data curation; Supervision. **Kevin S McIver**: Supervision. **Emanuel Hanski**: Conceptualization; Formal analysis; Supervision; Funding acquisition; Writing—original draft.

Source data underlying figure panels in this paper may have individual authorship assigned. Where available, figure panel/source data authorship is listed in the following database record: biostudies:S-SCDT-10_1038-S44319-025-00447-z.

## Disclosure and competing interests statement

The authors declare no competing interests.

# Expanded View Figures

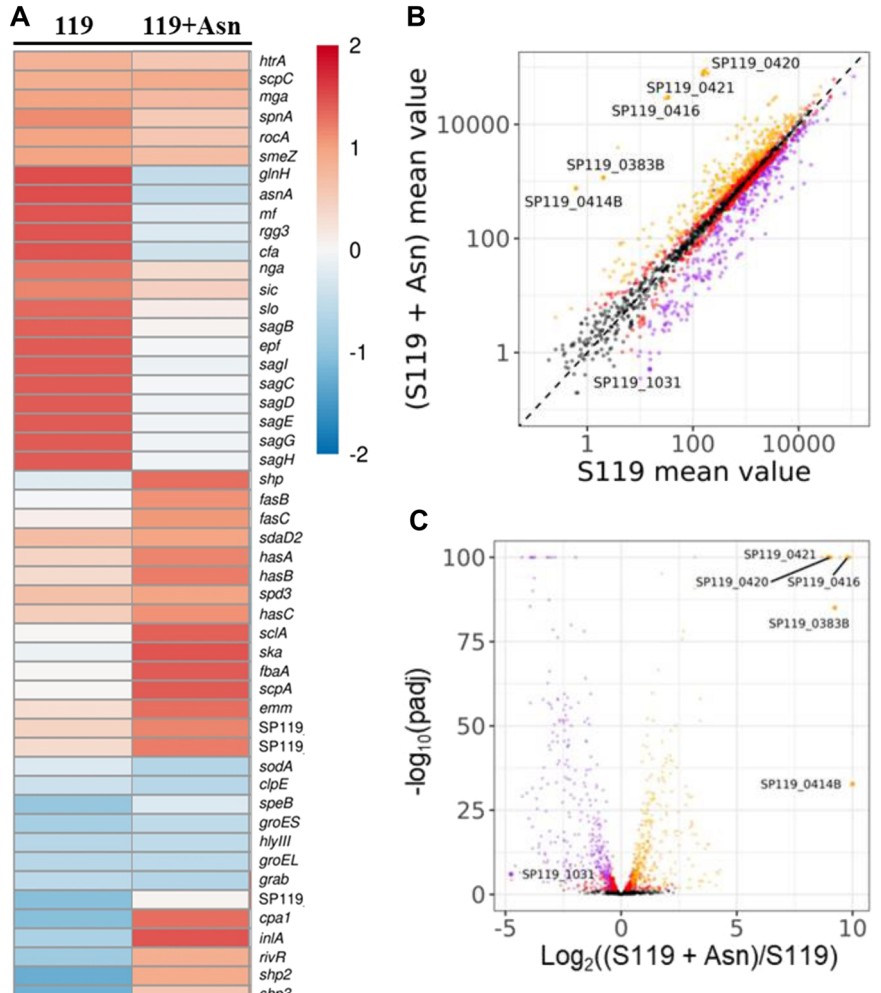

**Figure EV1. Asn regulates GAS transcriptome.**

(A) The heatmap represents the expression profile of genes, including virulence factors regulated by CovR/S directly or indirectly. (B, C) Visualization of mRNA-seq data. (B) A scatterplot matrix of the RNAseq data. Each dot represents a normalized mean value of the transcript number of the gene. (C) A Volcano plot of the same data set; each dot represents a gene with adjusted *P* < 0.05. as in Fig. 1. Data information: Four biological replicates were used (**A–C**).

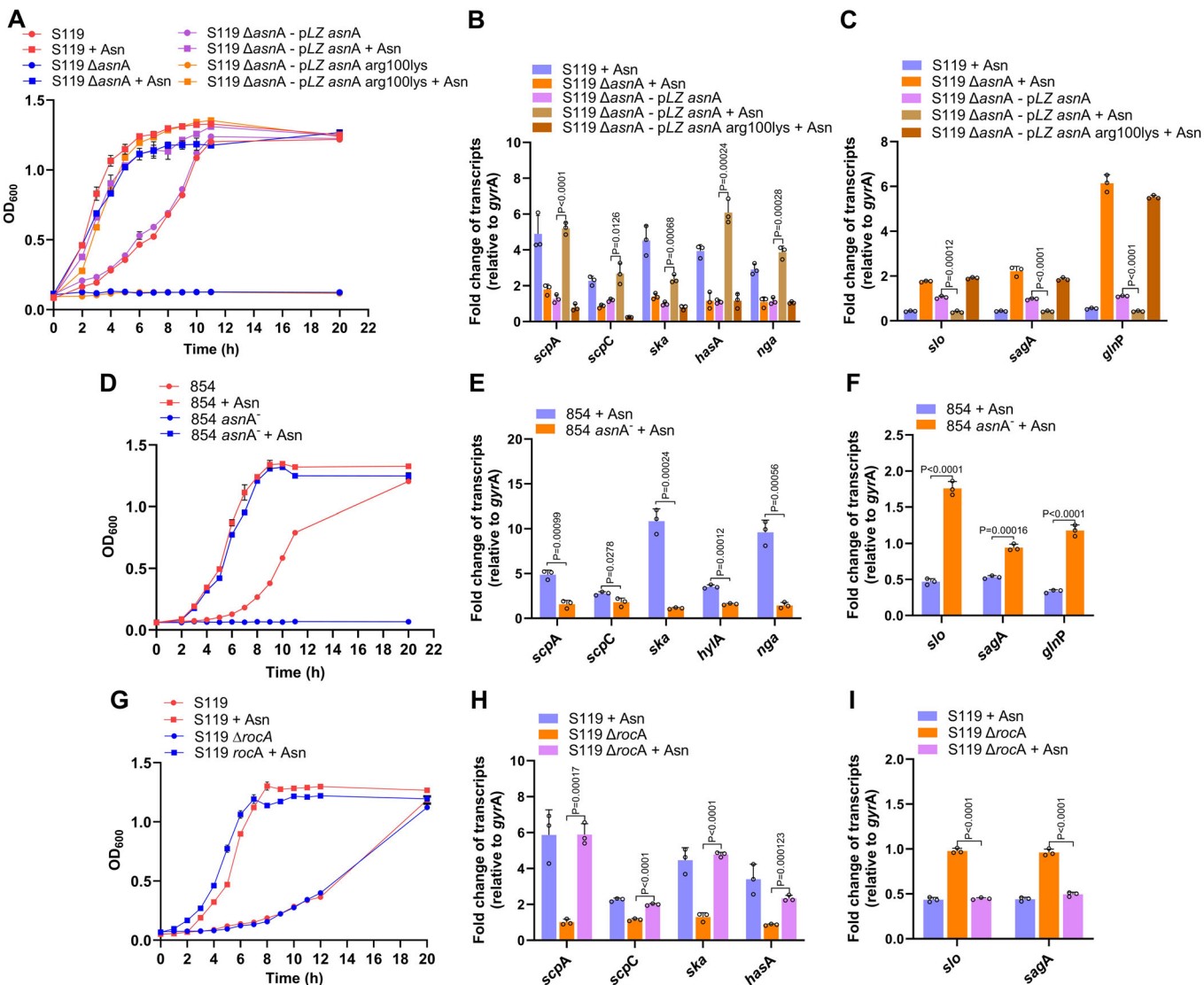

**Figure EV2. asnA is essential for Asn-mediated gene regulation.**

(A) The growth curves of stains S119, S119ΔasnA, S119ΔasnA-pLZasnA, and S119ΔasnA-pLZasnA arg100lys. (B, C) qRT-PCR determinations for Set 1 (B) and Set 2 (C) were conducted as described in Fig. 2. (D) The growth of the GAS 854 strain and its asnA⁻ derived mutant was monitored in CDM at indicated times. (E, F) qRT-PCR determinations for genes of Set 1 (E) and genes of Set 2 (F) were determined as described in Fig. 1. (H, I) RocA is not involved in Asn-mediated effects. The growth (G) and qRT-PCR determinations for Set 1 (H) slo and sagA, and (I) for S119 and its ΔrocA-derived mutant were performed as described above. Data information: Three biological replicates were used (A–I). The data shown represent the means ± SD. Statistical analysis was performed using an unpaired two-tailed t test (B, C, E, F, H, I). Source data are available online for this figure.

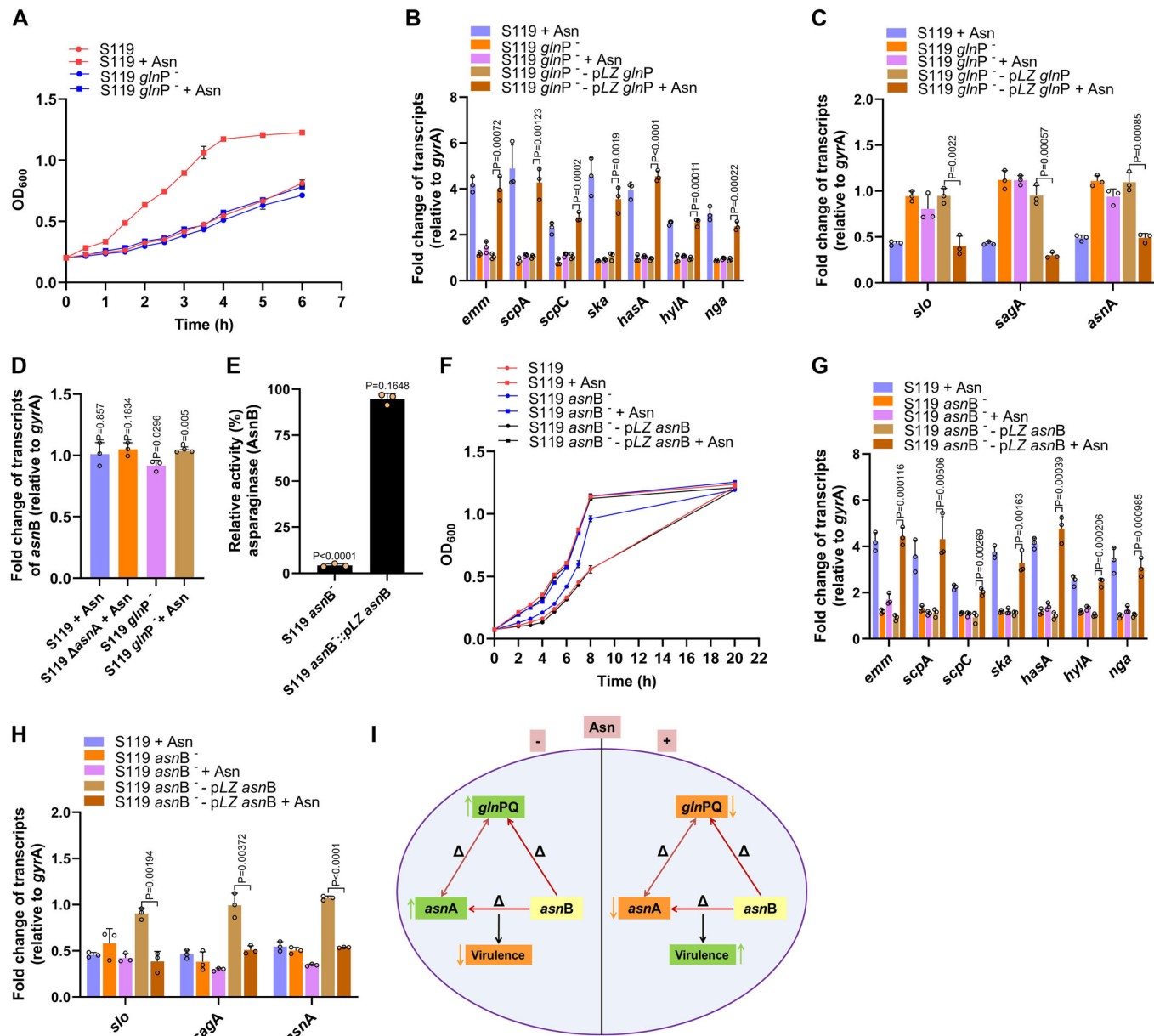

**Figure EV3. *gln*P and *asn*B are essential for Asn-mediated gene regulation.**

(A) The growth of S119 and S119 *gln*P⁻ in the absence or presence of Asn was determined (see Fig. 3B for the experimental details). (B, C) qRT-PCR determinations were conducted on Set 1 (B) or Set 2 of genes (C) using the indicated strains in CDM without or with Asn. (D) The *asn*B transcript is unaffected by the presence or absence of Asn or by the deficiency of AsnA or GlnPQ activities. qRT-PCR determinations of *asn*B transcript abundance were conducted as in (B, C). (E) Asparaginase activity (AsnB) was determined in the cell pellets of the indicated strains grown in THY. The asparaginase activity of WT S119 was used as a control. (F) The growth of the indicated strains was monitored in CDM in the absence or presence of Asn (10 and 100 μg ml⁻¹). (G and H) qRT-PCR determinations of Set 1 (G) and Set 2 (H), genes of the indicated strains grown in CDM without or with Asn. For all qRT-PCR data, fold change was calculated by comparing with normalized transcript abundance in the GAS S119 strain without Asn. (I) A graphical representation summarizes the orchestrated interplay between *asn*A, *gln*P, and *asn*B genes to maintain the balance of intracellular Asn in GAS. Data information: Three biological replicates were used (A–H). The data shown represent the means ± SD. Statistical analysis was performed using an unpaired two-tailed *t* test (B–E, G, H). Source data are available online for this figure.

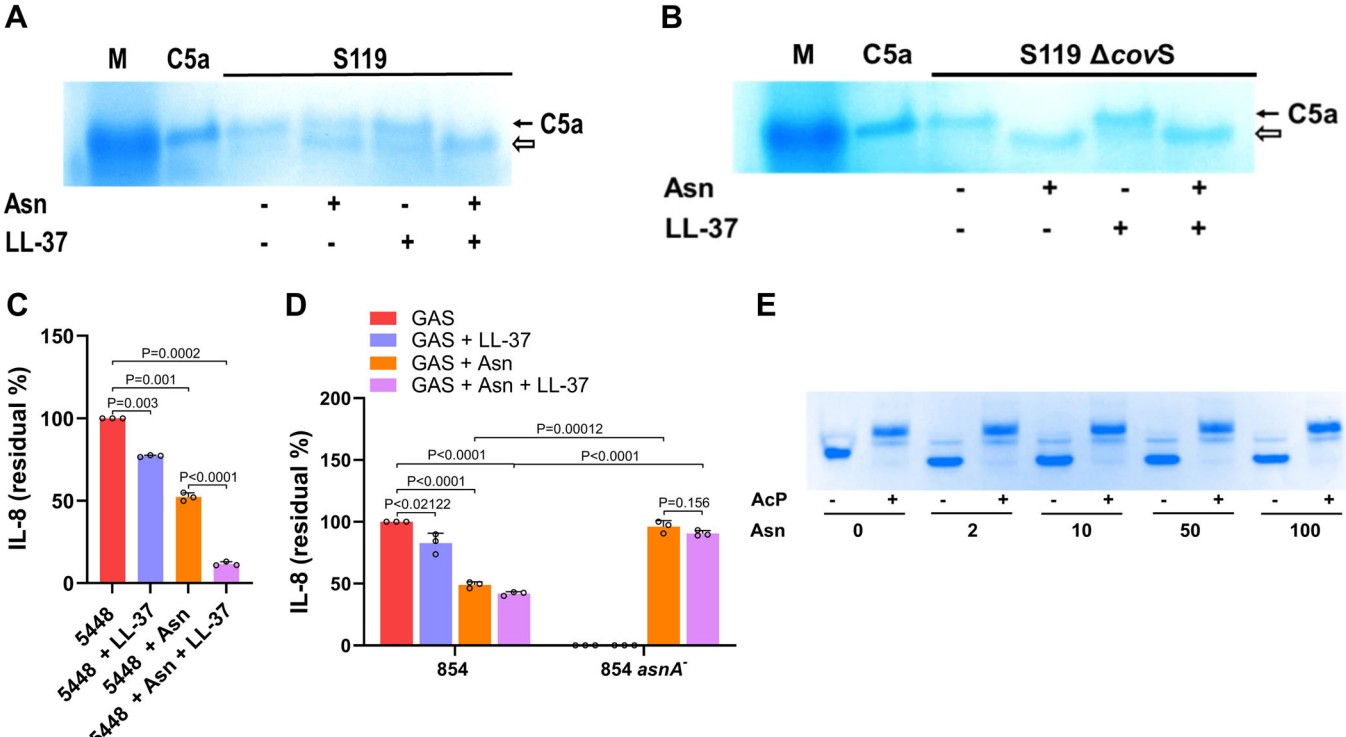

**Figure EV4. Functional assays and in-vitro phosphorylation of CovR.**

(A, B) Degradation of recombinant C5a by cultures supernatant containing the C5a-peptidase enzyme (ScpA). GAS was grown in CDM without or with Asn or/and LL-37. Supernatants of the indicated strains were incubated with recombinant human C5a, resolved on Tris-tricine gels, and visualized by Coomassie blue staining. M represents a marker, and an empty arrow is the cleaved C5a. The data are representative of two independent experiments. (C, D) Quantitation of IL-8 degradation by ScpC present in the supernatants of the indicated strains by ELISA. The IL-8 residual content in all supernatants was normalized to that of the GAS 5448 (C) and 854 (D) strain without Asn. (E) Asn does not affect in-vitro CovR phosphorylation. Purified CovR was incubated in the absence (−) and presence (+) of acetyl phosphate (Ac-P) as a phosphate donor and the indicated concentrations of Asn ($\mu g\ ml^{-1}$). The protein samples were resolved on Phos-tag SDS-PAGE gel and visualized. Data information: Three biological replicates were used (C, D). The data shown represent the means ± SD. Statistical analysis was performed using an unpaired two-tailed t test (C, D). Source data are available online for this figure.

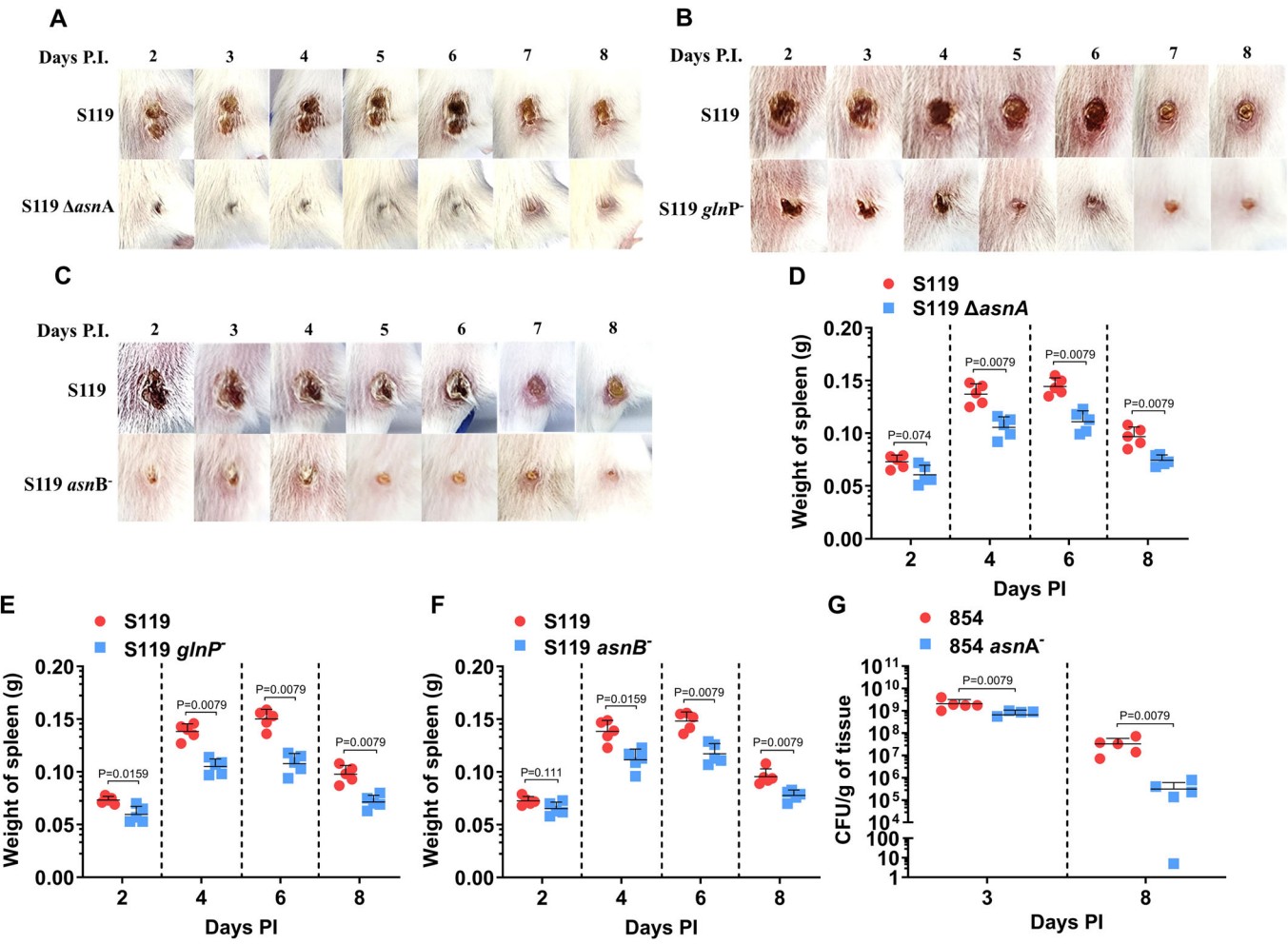

**Figure EV5.  *asn*A, *gln*P, and *asn*B mutants are attenuated in the sublethal murine model of human NF.**

(A–C) Mice were injected subcutaneously, and representative images of lesion progression at indicated time points after infection with S119 Δ*asn*A (A), S119 *gln*P⁻ (B), and S119 *asn*B⁻ (C), compared to WT S119 are shown. (D–F) Spleen weight of mice infected subcutaneously with S119 Δ*asn*A (D), S119 *gln*P⁻ (E), and S119 *asn*B⁻ (F), in comparison to S119, was determined. (D–G) The deletion of *asn*A produces an attenuated mutant in GAS strain 854. Mice were infected subcutaneously, and CFU counts per gram of soft tissue infected with 854 *asn*A⁻ compared to the wild-type, 854 were enumerated at indicated time intervals. Data information: Five mice per group per data point were used (D–G). The values shown represent the means ± SD. Statistical analysis was performed using the Mann–Whitney *U* test (D–G). Source data are available online for this figure.

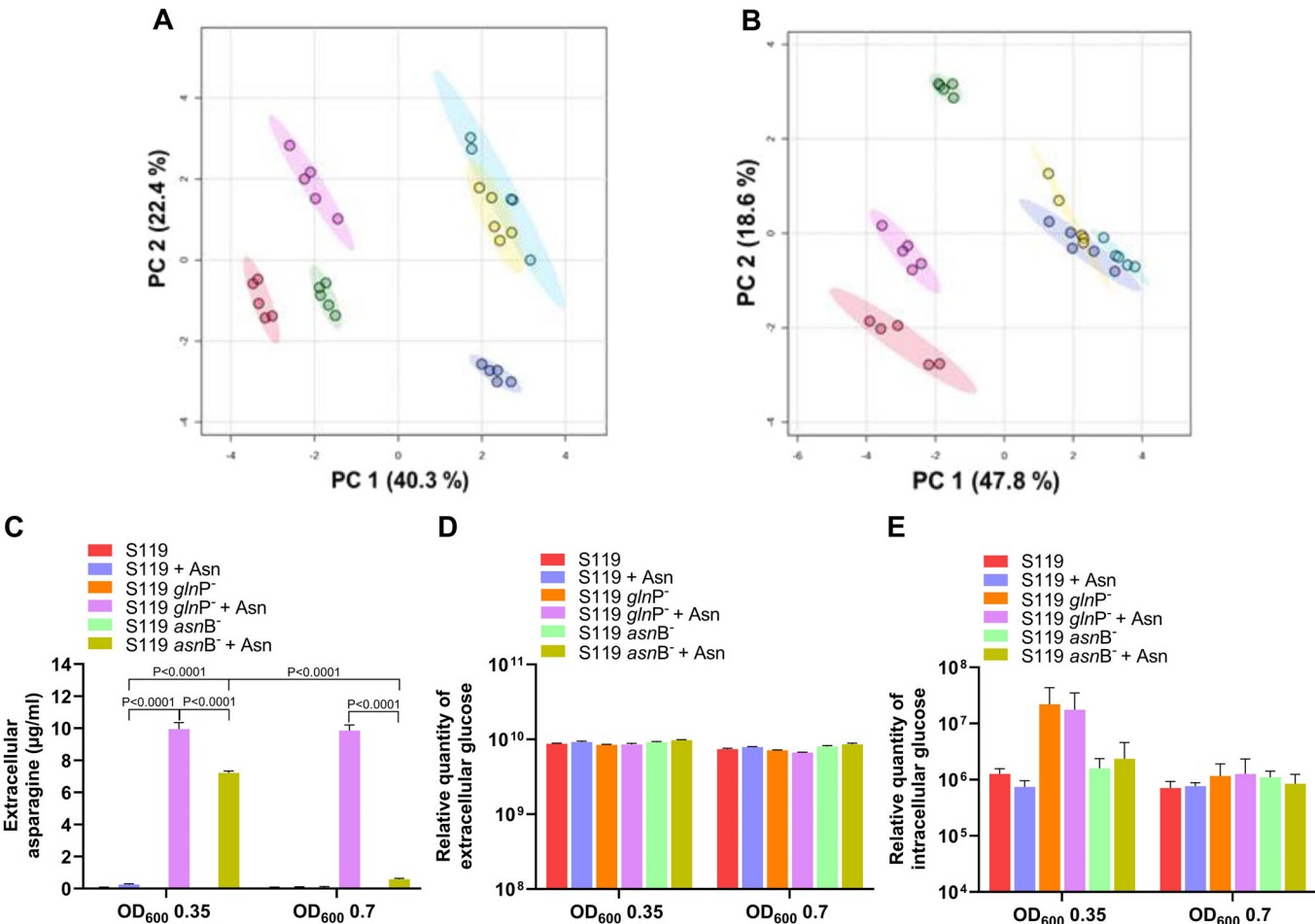

**Figure EV6. Coupling of metabolism to virulence.**

(A, B) The probabilistic principal component analyses (PCA) for the indicated pairs of strains grown in CDM or CDM supplemented with Asn to $OD_{600} = 0.35$ (A) or 0.7 (B). (C, D) Each circle represents one sample. Pink and yellow indicate S119 with and without Asn, respectively; blue and sky blue S119 *glnP⁻* with and without Asn; and red and green S119 *asnB⁻* with and without Asn. (C, D) Extracellular metabolite, Asn (C) content was determined for the indicated strains grown in CDM or CDM-supplemented with Asn to $OD_{600} = 0.35$ or 0.7. (D) The relative amount of glucose was determined for the indicated strains grown in CDM or CDM supplemented with Asn at $OD_{600} = 0.35$ or 0.7. (E) The relative amount of intracellular glucose was determined as above at $OD_{600} = 0.35$ or 0.7. Data information: Five biological replicates were used (A–E). The values shown represent the means ± SD. Statistical analysis was performed using an unpaired two-tailed *t* test (C–E). Source data are available online for this figure.

