## [Peer Review File · EMBO Reports]

Group A Streptococcal Asparagine Metabolism Regulates Bacterial Virulence

Emanuel Hanski, Abhinay Sharma, Aparna Anand, Miriam Ravins, Xiaolan Zhang, Nicola Horstmann, Samuel Shelburne, and Kevin McIver

Corresponding author(s): Emanuel Hanski (emanuelh@ekmd.huji.ac.il)

Review Timeline:	Transfer Date:	17th Dec 24
	Editorial Decision:	18th Dec 24
	Revision Received:	6th Feb 25
	Editorial Decision:	11th Mar 25
	Revision Received:	24th Mar 25
	Accepted:	1st Apr 25

Editor: Achim Breiling

Transaction Report: This manuscript was transferred to EMBO reports following peer review at The EMBO Journal.

Dear Prof. Hanski,

Thank you for transferring your manuscript to EMBO reports. I now went through the manuscript, the referee reports from The EMBO Journal (attached again) and your revision plan. The referees have several comments, concerns, and suggestions to improve the manuscript, indicating that a major revision of the manuscript is necessary to allow publication of the study.

Going through your revision plan, it seems that the referee points will be adequately addressed during revision to allow publication at EMBO reports. I thus invite you to revise your manuscript accordingly with the understanding that all concerns must be addressed in the revised manuscript and/or in a final detailed point-by-point response as indicated in your revision plan.

EMBO Reports emphasizes novel functional insight with clear in vivo relevance over detailed mechanistic insight. Thus, EMBO Reports will not require addressing points regarding more mechanism experimentally, but welcomes such data, in case you already have it. However, it will be necessary that during the revision you address all points questioning the main conclusions of the study, and all technical concerns, or points regarding the experimental designs, model systems used, or data presentation.

Acceptance of your manuscript will depend on a positive outcome of another round of review at EMBO reports, using the same referees.

1) a .docx formatted version of the final manuscript text (including legends for main figures, EV figures and tables), but without the figures included. Please make sure that changes are highlighted to be clearly visible. Figure legends should be compiled at the end of the manuscript text.

2) individual production quality figure files as .eps, .tif, .jpg (one file per figure), of main figures and EV figures. Please upload these as separate, individual files upon re-submission. Please make sure that all figure panels are called out separately and sequentially in the manuscript text

For more details please refer to our guide to authors:

See also our guide for figure preparation:

Moreover, please consult our guidelines for figure legend preparation:

4) a complete author checklist, which you can download from our author guidelines (<https://www.embopress.org/page/journal/14693178/authorguide>). Please insert page numbers in the checklist to indicate where the requested information can be found in the manuscript. The completed author checklist will also be part of the RPF.

Please also follow our guidelines for the use of living organisms, and the respective reporting guidelines:
<http://www.embopress.org/page/journal/14693178/authorguide#livingorganisms>

5) that primary datasets produced in this study (e.g. RNA-seq, ChIP-seq and array data) are deposited in an appropriate public database. This is now mandatory (like the COI statement). If no primary datasets have been deposited in any database, please state this in this section (e.g. 'No primary datasets have been generated and deposited').

The accession numbers and database should be listed in a formal "Data Availability" section (placed after Materials & Methods) that follows the model below. Please note that the Data Availability Section is restricted to new primary data that are part of this study.

Data availability

8) Regarding data quantification and statistics, please make sure that the number "n" for how many independent experiments were performed, their nature (biological versus technical replicates), the bars and error bars (e.g. SEM, SD) and the test used to calculate p-values is indicated in the respective figure legends (also for potential EV figures and all those in the final Appendix). Please also check that all the p-values are explained in the legend, and that these fit to those shown in the figure. Please provide statistical testing where applicable. Please avoid the phrase 'independent experiment', but clearly state if these were biological or technical replicates. Please also indicate (e.g. with n.s.) if testing was performed, but the differences are not significant. In case n=2, please show the data as separate datapoints without error bars and statistics.

See also:

<http://www.embopress.org/page/journal/14693178/authorguide#statisticalanalysis>

If n<5, please show single datapoints for diagrams. Please add to each legend (main, EV figures, Appendix, where applicable) a 'Data Information' section explaining the statistics used or providing information regarding replicates and scales. See: <https://www.embopress.org/page/journal/14693178/authorguide#figureformat>

9) Please add scale bars of similar style and thickness to any microscopic images, using clearly visible black or white bars (depending on the background). Please place these in the lower right corner of the images themselves. Please do not write on or near the bars in the image but define the size in the respective figure legend.

10) Please note our reference format:

11) We updated our journal's competing interests policy in January 2022 and request authors to consider both actual and perceived competing interests. Please review the policy <https://www.embopress.org/competing-interests> and add a statement declaring your competing interests. Please name that section 'Disclosure and Competing Interests Statement' and add it after the

author contributions section.

12) Please order the sections like this using these names:

Title page - Abstract - Keywords - Introduction - Results - Discussion - Methods - Data availability section (DAS) - Acknowledgements (including funding information) - Disclosure and Competing Interests Statement - References - Figure legends - Expanded View Figure legends

13) Please provide the abstract written in present tense throughout.

14) Please make sure that all the funding information is also entered into the online submission system and is complete and similar to the one in the manuscript text file (in the Acknowledgements).

15) We now use CRediT to specify the contributions of each author in the journal submission system. CRediT replaces the author contribution section. Please use the free text box to provide more detailed descriptions. Thus, please do NOT provide your final manuscript text file with an author contributions section. See also guide to authors: <https://www.embopress.org/page/journal/14693178/authorguide#authorshipguidelines>

16) All materials and methods used need to be described in the main text using our 'Structured Methods' format, which is required for all research articles. According to this format, the Methods section should include a Reagents and Tools Table (listing key reagents, experimental models, software, and relevant equipment and including their sources and relevant identifiers), uploaded as separate file, followed by a Methods section in which we encourage the authors to describe their methods using a step-by-step protocol format with bullet points, to facilitate the adoption of the methodologies across labs. More information on how to adhere to this format as well as downloadable templates (.doc or .xls) for the Reagents and Tools Table can be found in our author guidelines (section 'Structured Methods'):

I look forward to seeing a revised form of your manuscript when it is ready.

Yours sincerely,

Referee #1:

This manuscript investigates the role of *asnA* (which encodes asparagine synthetase) and its relationship with *glnPQ* and *asnB* (which encodes asparaginase) in mediating intracellular asparagine (Asn) levels. The findings indicate that Asn enhances metabolism and increases virulence in Group A *Streptococcus* (GAS), proposing a potential therapeutic target for GAS infections. Below are the comments.

Major comments:

1. There are major issues with the statistical analysis of the data in Figures 1-4 and S2-S3, as only two replicates were used, making it impossible to calculate standard deviation (SD). Therefore, error bars have to be removed. I am not sure how this major error could occur in this manuscript. Importantly, the authors must ensure that all results are based on three independent measurements. This issue must be addressed, as significant differences cannot be determined otherwise.
2. Figure 1B and lines 139-142: The significantly perturbed genes should be labeled in Figure 1B. For example, transcriptional changes in the virulence-associated genes such as *emm*, *scpA*, *scpC*, and *hasA* should also be presented. It is unknown if all

these genes were differentially expressed. Were there other virulence-associated genes significantly perturbed in Figure 1B? Did the authors examine their expression levels across different mutants and conditions?

3. The authors used OD values to normalise the metabolomic data across different samples. This approach is based on the assumption that the cell sizes of different mutants are comparable to that of the wild-type strain. This assumption should be confirmed with microscopy results.

4. Lines 393-395: The authors discuss the relative quantity of intracellular glucose only at OD=0.7, without mentioning the corresponding values at OD=0.35. Notably, the relative quantity of intracellular glucose substantially increased in the *glnP* mutant at OD=0.35 with or without Asn (nearly a 10-fold increase compared to the other four strains, Appendix Fig. S6E). Did this increase compensate for the decreased Asn levels? When determining the relative quantity of intracellular glucose, were any quality controls used to show acceptable variability of the extraction method across different samples? The authors should provide the raw LC/MS data for these values.

Other comments:

1. The references cited in the Introduction are outdated. For example, the first reference was published in 2013 which is 11 years ago. The authors should also cite the latest papers whenever possible.

2. The last paragraph of the Introduction should be shortened and made more concise.

3. Lines 118-119: The authors tested Asn concentrations ranging from 0.5 to 10 µg/mL, noting that bacterial growth peaked at 10 µg/mL. Did the authors examine concentrations higher than 10 µg/mL? If so, what effects were observed?

4. Lines 161-164 and Figures S1C-D: Could the authors clarify why they opted for knockout of *covS* while insertional inactivation for *covR*? This inconsistency also appears with other genes, such as *asnA* (KO) and *asnB* (insertion).

5. Figures 2B-C: The authors should include the qRT-PCR data for S119Δ*asnA*.

6. Lines 316-318: This sentence references Figure 4G-H.

7. Lines 364-366: Figure S6C should be cited here.

8. Figure 6A: It is surprising that intracellular Asn remains at such a high level in the *asnB* mutant in the presence of Asn. Due to *AsnB* dysfunction, intracellular Asn cannot be degraded, leading to intracellular accumulation. However, over the time, bacterial cells might be able to export excess Asn to maintain homeostasis. Did the authors check the expression of Asn-related exporters? Were they upregulated in the *AsnB* mutant?

Referee #2:

The authors report to describe a new mechanism by which asparagine increases GAS virulence, growth and metabolic rate by decreasing the levels of intracellular ATP and therefore reducing the phosphorylation and activity of the *CovR* regulator. Despite showing reduced ATP levels in the presence of asparagine it is not clear to me how asparagine controls ATP levels.

This work includes a large amount of knock out mutants and different conditions tested and the authors could show several findings. However how important each finding per se is, how these findings fit into the story and how they should be concatenated is very unclear. The interesting fact that ATP levels control GAS virulence, growth and metabolism mirrors what happens in cancer cells growth. However this was also not explained in a very clear way and possible strategies using GAS targets to treat disease are poorly outlined.

The manuscript is very dense and difficult to read, most information gets lost as findings are not presented in a very clear and linear way. It is difficult to grasp the significance of the various findings and how they are related to each other. The quality of the English is not ideal. The manuscript would profit if it were reorganized both scientifically and structurally to convey the message in a clearer way.

Referee #3:

The study investigates how asparagine (Asn) metabolism influences the virulence of Group A *Streptococcus*, a pathogen responsible for a wide range of infections. The authors report that that Asn in the growth medium is modulating transcriptome, *CovR* phosphorylation, and GAS virulence. Mutations affecting Asn synthesis, import, or degradation disrupt Asn balance and reduce virulence in a murine model. The study is in an important subject area, the results interesting, and I think it can become strong candidate for *Embo J*. However I have a few major comments, when it comes to causality and the interpretation of the results.

Major points:

Results und Result interpretation, related to Figure 1:

The experiments are well-executed; however, I find the authors' explanation that transcriptomic differences are caused by the "absence or presence" of Asn somewhat problematic. While Asn is either absent or present in the growth media, within the cell, Asn is always present-either through uptake (when supplemented) or self-synthesis. Strictly speaking, the observed differences are not caused by the "absence or presence" of Asn but rather correlate with the activity of the biosynthetic pathway, the uptake

of Asn, or a combination of both. This point necessitates a substantial revision of the first section and the abstract.

Results related to Figure 2:

The results here are intriguing, but I take issue with the interpretation. It is not valid to assume that Asn synthesis remains active when Asn is supplemented and taken up. Under these conditions, the most likely function of AsnA must be a different one to Asn biosynthesis; perhaps Asn catabolism. A metabolic labeling or isotope tracing experiment could help clarify this point.

Results related to Figure 5:

The results make sense overall, but the increased clearance rates are more accurately described as correlating with reduced Asn uptake. While it is plausible that this results from Asn-induced transcriptomic changes and other downstream responses, alternative explanations cannot be ruled out: Strictly speaking, the problem is that these cells do not just see Asn; host cells provide a broad range of metabolites, not just Asn, and metabolic pathways along with efflux mechanisms are highly interconnected. It would be important to include a caveat in the discussion, as causality seems not robustly established here.

The role of ATP in the downstream cellular responses from switching from Asn biosynthesis to uptake

Here I see my biggest problem with the interpretation of the results. The abstract and some parts of the discussion state or imply that Asn metabolism acts directly via ATP, which seems to overstate the findings. The results shown in the manuscript are consistent with Asn catabolism feeding into energy metabolism-an entirely plausible outcome since the carbon backbone of every glucogenic amino acid is channeled into carbon catabolism, ultimately leading to ATP production. However, this does not demonstrate that changes in ATP levels are causally responsible for altered virulence or gene expression changes. I strongly recommend tempering statements, particularly in the abstract and concluding statements, implying such causality : the metabolic shifts resulting from switching between Asn biosynthesis and uptake likely involve hundreds of metabolites, with ATP being only one of them.

A point-by-point response to reviewer comments:

Reviewer 1

1. There are major issues with the statistical analysis of the data, making it impossible to calculate standard deviation (SD). Therefore, error bars have to be removed. I am not sure how this major error could occur in this manuscript. Importantly, the authors must ensure that all results are based on three independent measurements. This issue must be addressed, as significant differences cannot be determined otherwise.

Answer: We performed the experiments to obtain another biological replication and calculated the standard deviation (SD) values.

2. Figure 1B and lines 139-142: The significantly perturbed genes should be labeled in Figure 1B. For example, transcriptional changes in the virulence-associated genes such as *emm*, *scpA*, *scpC*, and *hasA* should also be presented. It is unknown if all these genes were differentially expressed. Were there other virulence-associated genes significantly perturbed in Figure 1B? Did the authors examine their expression levels across different mutants and conditions?

Answer: We incorporated another Heatmap describing the changes in primary CovR/S-regulated genes, and the raw data is available in GEO accession number GSE268517. We mentioned in the text that out of the total number of genes (n=1839) detected in RNAseq, a total of 27.35% (n=503) of genes were affected; out of these affected genes, 13.97% (n=257) genes were upregulated, and 13.37% (n=246) were downregulated by at least 2 fold in the treatment condition (+Asn) in GAS S119 strain. We performed RT-PCR validations on the selected genes known to be regulated by CovR. We performed RT-PCR determination for each mutant on the same set of genes except 2. As the general comment (A) indicates, phosphorylated CovR negatively regulates most virulent genes. Thus, minor dephosphorylation level changes should significantly impact virulence gene expression. However, the notion that CovR-P switches on and off state across **all genes at all**

conditions is naïve. Infinitesimal CovR-P changes may differentially affect gene expression. Thus, the core of genes chosen for the mutant in this study is representative and entirely legitimate.

3. The authors used OD values to normalise the metabolomic data across different samples. This approach is based on the assumption that the cell sizes of different mutants are comparable to that of the wild-type strain. This assumption should be confirmed with microscopy results.

Answer: We wish to clarify that metabolic data across different samples were normalized by protein amount in each sample (this has been explicitly explained in the Methodology section lines 584-586). We are aware that OD values may not represent the actual number of bacteria; therefore, we normalized the values to protein content.

4. Lines 393-395: The authors discuss the relative quantity of intracellular glucose only at OD=0.7, without mentioning the corresponding values at OD=0.35. Notably, the relative quantity of intracellular glucose substantially increased in the *glnP* mutant at OD=0.35 with or without Asn (nearly a 10-fold increase compared to the other four strains, Appendix Fig. S6E). Did this increase compensate for the decreased Asn levels? When determining the relative quantity of intracellular glucose, were any quality controls used to show acceptable variability of the extraction method across different samples? The authors should provide the raw LC/MS data for these values.

Answer: Thank you for your concern. We aimed to show the extracellular and intracellular glucose levels to confirm whether glucose as a carbon source for energy generation was a limiting factor of bacterial growth. We found that glucose was amply present under all conditions of GAS strains and mutants, and ATP was not depleted because of the scarcity of glucose.

As already indicated above, we normalized to the protein level of the extract and not OD, as indicated above. As suggested, we show the relative quantity of intracellular glucose at OD 0.35 and 0.7 (Fig. EV6D).

The reasons for the higher glucose levels in *glnPQ* mutant at OD 0.35 are unknown. One can speculate that the uptake of Asn from the host at early

growth stages significantly affects the rate of glycolysis. Therefore, the *glnPQ* mutant accumulates glucose more efficiently. Alternatively, the lack of Asn uptake may affect glucose uptake, consumption, and other scenarios. Further studies are required to answer this question. As indicated above, the data was normalized to protein levels, not OD.

Raw data will be provided.

Other comments:

1. The references cited in the Introduction are outdated. For example, the first reference was published in 2013 which is 11 years ago. The authors should also cite the latest papers whenever possible.

Answer: A more recent reference was used.

2. The last paragraph of the Introduction should be shortened and made more concise.

Answer: We shortened the last paragraph.

3. Lines 118-119: The authors tested Asn concentrations ranging from 0.5 to 10 $\mu\text{g/mL}$, noting that bacterial growth peaked at 10 $\mu\text{g/mL}$. Did the authors examine concentrations higher than 10 $\mu\text{g/mL}$? If so, what effects were observed?

Answer: We also tested the bacterial growth at 100 $\mu\text{g/mL}$ concentration Asn. There was some increase in the growth rate, but not significant (it is mentioned in the text: line 115 and Figure 3A). To prevent an overwhelming flood of the system by extracellular Asn, we decided to use 10 $\mu\text{g/mL}$ of Asn.

4. Lines 161-164 and Figures S1C-D: Could the authors clarify why they opted for knockout of *covS* while insertional inactivation for *covR*? This inconsistency also appears with other genes, such as *asnA* (KO) and *asnB* (insertion).

Answer: Our purpose was to stop the functionality of *covS*, *covR*, *asnA*, and *asnB* to confirm their activity roles in Asn metabolism and GAS pathogenesis. Both knockouts and insertional inactivation are legitimate for this purpose if there is no reversal of the genetic phenotype during the test. It was verified by growing the bacteria with and without the corresponding antibiotics and counting CFU. Most importantly, when complemented by plasmids harboring intact genes, the mutants reversed their phenotype to that of the WT (shown for each mutant), **which unequivocally confirms their validity.**

5. Figures 2B-C: The authors should include the qRT-PCR data for S119 Δ *asnA*.

Answer: We have already shown in Figure 2A that S119 Δ *asnA* does not grow without Asn, so qRT-PCR cannot be performed.

6. Lines 316-318: This sentence references Figure 4G-H.

Answer: It was corrected.

7. Lines 364-366: Figure S6C should be cited here.

Answer: The figure was cited.

8. Figure 6A: It is surprising that intracellular Asn remains at such a high level in the *asnB* mutant in the presence of Asn. Due to *AsnB* dysfunction, intracellular Asn cannot be degraded, leading to intracellular accumulation. However, over the time, bacterial cells might be able to export excess Asn to maintain homeostasis. Did the authors check the expression of Asn-related exporters? Were they upregulated in the *AsnB* mutant?

Answer: We were also surprised by the finding that Asn was not detected (not even at OD= 0.7) in the extracellular medium when the bacteria were grown in CDM without Asn (see Figure EC6C), although *AsnA* forms Asn. This finding

indicates that GAS is not exporting Asn outside the cell under these conditions, although the AsnA synthesizes it during the growth. We clearly show this for the WT strain. We did not test it for the *asnB* mutant.

Referee#2:

The authors report to describe a new mechanism by which asparagine increases GAS virulence, growth and metabolic rate by decreasing the levels of intracellular ATP and therefore reducing the phosphorylation and activity of the CovR regulator. Despite showing reduced ATP levels in the presence of asparagine it is not clear to me how asparagine controls ATP levels.

These findings present the main essence of our manuscript. As the reviewer knows, GAS is a lactic bacterium that produces ATP only due to the glycolytic pathway and does not use the citric pathway since it lacks most of the enzymes required. The net gain in glycolysis is two molecules of ATP formed for one molecule of glucose consumed. If the uptake of Asn from the host by GlnPQ requires ATP hydrolysis, the same is true for Asn formation by AsnA, and both processes occur during growth; the net ATP level should be low under these conditions. The same scenario can be seen for cancerous cells due to the Warburg effect (they get their energy primarily from fermentation) or multiply under anaerobic conditions. Under these conditions, the ATP level is low, and Asn controls cell multiplication due to its signaling to mTOR1 (please see Krall *et al.*, 2021).

This work includes a large amount of knock out mutants and different conditions tested and the authors could show several findings. However how important each finding per se is, how these findings fit into the story and how they should be concatenated is very unclear. The interesting fact that ATP levels control GAS virulence, growth and Metabolism mirrors what happens in cancer cells growth. However this was also not explained in a very clear way and possible strategies using GAS targets to treat disease are poorly outlined.

Three enzymes determine the Asn bacterial concentration: QlnPQ, the ABC transporter that we identified here to export Asn from the host; AsnA, which synthesizes Asn from aspartate and ammonia; and AsnB, which degrades Asn to aspartate and ammonia. Therefore, we had to construct all these mutants to test if the observations we see *ex-vivo* in the bacteria grown in CDM are relevant to the *in-vivo* data in the animal model. We made a genuine effort to explain this better in the revised version.

Strategies using GAS targets to treat disease are poorly outlined.

Asparaginase is a drug used to treat acute lymphoblastic leukemia (ALL) (in the last 40 years) and is being studied in the treatment of some other types of cancer. We showed in 2014 that it treats GAS infections (Baruch et al. 2014).

"PERK induces resistance to cell death elicited by endoplasmic reticulum stress and chemotherapy". IC Salaroglio, E Panada, E Moiso, I Buondonno... - Molecular cancer, 2017. We used novel PERK inhibitors to demonstrate that they protect the host against invasive GAS infection (Anand et al. 2021).

Several groups are trying to develop anti-AsnA drugs to treat cancer (for example, please see: "Metabolism of asparagine in the physiological state and cancer". Q Yuan, L Yin, J He, Q Zeng, Y Liang, Y Shen... - Cell Communication and ..., 2024)

The manuscript is very dense and difficult to read, most information gets lost as findings are not presented in a very clear and linear way. It is difficult to grasp the significance of the various findings and how they are related to each other. The quality of the English is not ideal. The manuscript would profit if it were reorganized both scientifically and structurally to convey the message in a clearer way.

.

We made a genuine effort to explain this better in the revised version.

Referee# 3:

The study investigates how asparagine (Asn) metabolism influences the virulence of Group A Streptococcus, a pathogen responsible for a wide range of infections. The authors report that that Asn in the growth medium is modulating transcriptome, CovR phosphorylation, and GAS virulence. Mutations affecting Asn synthesis, import, or degradation disrupt Asn balance and reduce virulence in a murine model. The study is a in an important subject area, the results interesting, and I think it can become strong candidate for Embo J. However I have a few major comments, when it comes to causality and the interpretation of the results.

Major points Results and Result interpretation, related to Figure 1:

The experiments are well-executed; however, I find the authors' explanation that transcriptomic differences are caused by the "absence or presence" of Asn somewhat problematic. While Asn is either absent or present in the growth media, within the cell, Asn is always present through uptake (when supplemented) or self-synthesis. Strictly speaking, the observed differences are not caused by the "absence or presence" of Asn but rather correlate with the activity of the biosynthetic pathway, the uptake of Asn, or a combination of both. This point necessitates a substantial revision of the first section and the abstract.

Answer: We partially agree with the comment that GAS can synthesize Asn inside cells, but this Asn is insufficient to increase the growth rate seen when Asn is added to the CDM. Furthermore, we found that the transcription of some of the replication genes is also affected by Asn uptake (see our previous paper Ref Baruch et al., 2014 Cell) and the current manuscript. We show that Asn presence upregulated *polA*, *lig*, and *danX*. Thus, it appears that the intracellular level of Asn has to cross some threshold to affect gene

transcription and growth rate, but the increase in Asn is limited because when it is too high in the AsnB mutant, the mutant loses the transcriptional increase and is attenuated *in vivo*. We understand the difficulty pointed out by the reviewer and modified the text accordingly to explain this notion better.

Results related to Figure 2: The results here are intriguing, but I take issue with the interpretation. It is not valid to assume that Asn synthesis remains active when Asn is supplemented and taken up. Under these conditions, the most likely function of AsnA must be a different one to Asn biosynthesis; perhaps Asn catabolism. A metabolic labeling or isotope tracing experiment could help clarify this point.

Answer: AsnA remains active even if we provide Asn from outside because we observed in qRT-PCR that the *asnA* gene also expresses in Asn-supplemented conditions, although the expression level goes down. AsnB plays an essential role in Asn catabolism but is not regulated by Asn since it is of type b having very high K_{cat} . All these results are provided.

To address this issue further, we also determined the level of aspartate, which serves as a substrate for AsnA and is formed when AsnB degrades Asn to aspartate and ammonia. Here are the results. So, we do not think that AsnA works in catabolism.

Results related to Figure 5: The results make sense overall, but the increased clearance rates are more accurately described as correlating with reduced Asn uptake. While it is plausible that this results from Asn-induced transcriptomic changes and other downstream responses, alternative explanations cannot be ruled out: Strictly speaking, the problem is that these cells do not just see Asn; host cells provide a broad range of metabolites, not just Asn, and metabolic pathways along with efflux mechanisms are highly interconnected. It would be important to include a caveat in the discussion, as causality seems not robustly established here.

Indeed, we do not know what happens to GAS metabolism *in vivo* during the infection, in which GAS passes through so many different niches of the host. Nevertheless, we demonstrated that Asn uptake is critical because when blocked by adding Kidrolase (asparaginase from Gram-negative bacteria, a drug that treats acute lymphoblastic leukemia), it prevents GAS infection in the mouse model (Baruch et al., 2014). Moreover, we showed that treating mice with PERK inhibitors inhibits the formation of Asn in the GAS-infected host cells and treats against invasive GAS disease (Anand et al., 2021).

The role of ATP in the downstream cellular responses from switching from Asn biosynthesis to uptake. Here I see my biggest problem with the interpretation of the results. The abstract and some parts of the discussion state or imply that the Asn metabolism acts directly via ATP, which seems to overstate the findings. The results shown in the manuscript are consistent with Asn catabolism feeding into energy metabolism-an entirely plausible outcome since the carbon backbone of every glucogenic amino acid is channeled into carbon catabolism, ultimately leading to ATP production. However, this does not demonstrate that changes in ATP levels are causally responsible for altered virulence or gene expression changes. I strongly recommend tempering statements, particularly in the abstract and concluding statements, implying such causality: the metabolic shifts resulting from switching between

Asn biosynthesis and uptake likely involve hundreds of metabolites, with ATP being only one of them.

We added Figure 6D, which shows that the ratio of ADP to ATP is significantly higher when Asn is added to the culture medium. Since ADP upregulates the phosphatase activity in structurally related TCS such as EnvZ/OmpR and PhoP/PhoQ, it is undoubtedly possible to assume that CovR is dephosphorylated when ATP is consumed and ADP is in excess.

Dear Prof. Hanski

Thank you for the submission of your revised manuscript to our editorial offices. I have now received the reports from two of the three referees that were asked to re-evaluate the study, you will find below. Original referee #1 was completely unresponsive to my invitations to re-assess the study.

As you will see, original referee #3 (now referee #2) supports the publication of the study in EMBO reports. However, referee #1 (original referee #2) has several remaining concerns and suggestions to improve the study, I ask you to address in a final revised manuscript. Please also provide a final p-b-p-response regarding the remaining points of the referee.

Moreover, I have these editorial requests I also ask you to address:

- Please add up to five keywords to the manuscript and order the sections like this, using these names:

Title page - Abstract - Keywords - Introduction - Results - Discussion - Methods - Data availability section - Acknowledgements (including the funding information) - Disclosure and Competing Interests Statement - References - Figure legends - Expanded View Figure legends

- Please remove the 'Supplementary Material' section. There is a Dataset EV1 mentioned, but not uploaded. Please check.

- Please remove the sentence 'The source data of this paper are collected in the following database record:' from the Data availability section.

- We now use CRediT to specify the contributions of each author in the journal submission system. CRediT replaces the author contribution section. Please use the free text box to provide more detailed descriptions and do NOT provide your final manuscript text file with an author contributions section. See also our guide to authors: <https://www.embopress.org/page/journal/14693178/authorguide#authorshipguidelines>

- Please upload a complete author checklist with your revision, which you can download from our author guidelines (<https://www.embopress.org/page/journal/14693178/authorguide>). Please insert page numbers in the checklist to indicate where the requested information can be found in the manuscript. The completed author checklist will also be part of the RPF.

- Please move the two Appendix Tables (bacterial strains and primer information) to the Reagents and Tools tables. Please update their callouts (see 'Reagents and Tools Table') and make sure that the tools table is called out where appropriate in the Methods section. Moreover, please upload the Reagents and Tools table as separate file and remove the table from the main manuscript text file. More information on how to adhere to this format as well as downloadable templates (.doc) for the Reagents and Tools Table can be found in our author guidelines (section 'Structured Methods'):

- Please move all the methods information and related references to the main manuscript text file. We do not allow supplementary methods. Please also remove the callouts to "Supplementary Materials". Finally, please delete the Appendix file.

- The separate Word file with EV figures and legends is not needed (EV figures are already uploaded separately and their legends are in the main manuscript). Please remove this.

- Please check again that the number "n" for how many independent experiments were performed, their nature (biological versus technical replicates), the bars and error bars (e.g. SEM, SD) and the test used to calculate p-values is indicated in the respective figure legends. Please also check that all the p-values are explained in the legend, and that these fit to those shown in the figure. Please provide statistical testing where applicable. Please avoid the phrase 'independent experiment', but clearly state if these were biological or technical replicates. Please also indicate (e.g. with n.s.) if testing was performed, but the differences are not significant. In case n=2, please show the data as separate datapoints without error bars and statistics. See also:

<http://www.embopress.org/page/journal/14693178/authorguide#statisticalanalysis>

If n<5, please show single datapoints for diagrams. Presently, most diagrams show no statistics! Please add this as indicated above. Moreover:

- Please note that n=2 in figures 4F. Please remove that stats (see above).

- Please note that the error bars are not defined in the legends of figures 3C-F.

- Please define the annotated p values ****/**/* as well as provide the exact p-values for the same in the legend of figure 2B, C; EV2 A-I; EV4 C, D; as appropriate.

- Please note that the exact p values are not provided in the legends of figures 1E, F; 3B, 4C, D, H; 5A-F; 6A-D; EV1 D, E; EV5 D-J; EV6 C

- Please indicate the statistical test used for data analysis in the legends of figures EV4 C, D

- Please add to each legend (main, EV and Appendix figures, where applicable) a 'Data Information' section explaining the statistics used or providing information regarding replicates and scales. See:

- Please make sure that all the funding information is also entered into the online submission system and that it is complete and similar to the one in the acknowledgement section of the manuscript text file. Presently, the grant from the Israeli Ministry of Innovation, Science and Technology grant (Grant number 0005663) is missing in the submission system. Please check.

- Thank you for providing the source data. Please upload the SD as one folder per figure, grouping together separate excel files for all panels for one figure (and ZIPed together). Please upload the SD for the EV figures as one folder containing sub-folders for each EV figures grouping together separate excel files for the panels for each EV figure.

In addition, I would need from you uploaded separately:

- a short, two-sentence summary of the manuscript (not more than 35 words).

- two to four short (!) bullet points highlighting the key findings of your study (two lines each).

- a schematic summary figure as separate file that provides a sketch of the major findings (not a data image) in jpeg or tiff format (with the exact width of 550 pixels and a height of not more than 400 pixels) that can be used as a visual synopsis on our website.

Best,

Referee #1:

Group A Streptococcal Asparagine Metabolism Regulates Bacterial Virulence

The authors investigate the role of asparagine (Asn) in GAS virulence expression. They could show that Asn improves GAS growth in minimal medium, enhances the expression of several virulence factors (emm, ScpA, ScpC, Ska, HasA, HylA, Nga) and reduces the expression of the asparagine synthetase AsnA and the asparagine transporter GlnP as well as of SLO and SLS, two streptolysins involved in scavenging Asn from eukaryotic cells, in a negative feedback loop. The authors show that CovS is required for Asn-mediated expression regulation of all genes examined, while CovR is only necessary for Asn-mediated regulation of virulence factors but not for AsnA and GlnP. They subsequently explore the role of AsnA and GlnP, both necessary for expression regulation of all genes tested. The asparaginase AsnB was only necessary for virulence factors upregulation. They show that Asn increases virulence factors activity by lowering intracellular ATP levels and therefore reducing CovR phosphorylation. All genes involved in Asn metabolism (AsnA, GlnP and AsnB) are necessary to enhance gas virulence in a murine necrotizing fasciitis mouse model.

General comments:

I am not sure the authors can claim that Asn metabolism regulates VFs in GAS, it seems more regulated by the presence/absence of Asn, rather than its metabolism. Please comment on that extensively (see also my comments below).

The manuscript quality is partly poorly written and should be made more concise for clarity. The text is embedded with many imprecisions, both graphical and grammatical, such as figures incorrectly labeled, statistics on figures missing, sentences missing subject/verb, just to mention some.

The titles in the result section should be more explanatory, describing the main finding of the paragraph and reflecting the title of the figure legends.

Abstract:

There is no mention of Asn-mediated gene expression regulation in the abstract, which I is interesting since these data cover about half of the findings. This information needs to be embedded in the abstract in a seamless way. A summary of the

manuscript may be better described at the end of the introduction, for instance (line 89-103).

Line 31: Human manifestations? I'd change to human diseases.

Lines 32-35: "Here, we show that asparagine (Asn) homeostasis in GAS is maintained by asparagine synthetase (AsnA), the ABC transporter (GlnPQ) that imports extracellular Asn, and a highly efficient asparaginase (AsnB) that degrades intracellular Asn". This is presented as a main finding in the abstract but this does not reflect what's in the manuscript. Additionally, I also fear the authors can't claim that they discover how Asn homeostasis is maintained in GAS since the function of all genes they cite is already known and they are all already known to be involved in Asn homeostasis. Please comment on this and change the abstract accordingly.

Introduction:

There is no description of the mechanisms of Asn-mediated tumor growth in the introduction that needs to be added. The authors suggest a similarity but do not describe what happens in tumor cells.

Line 55: Bacterial infections or generally?

Line 66: It was repeatedly shown that GAS is mainly an extracellular pathogen but can also survive intracellularly, this has to be specified here.

Lines 92-93: "Furthermore, we found that the ABC transporter GlnPQ is responsible for importing Asn...". Is this not already known from the literature? Please comment!

Results:

As already mentioned, the titles in the result sections should be more explanatory, describing the main finding of the paragraph. Please change them accordingly!

Lines 107-108: Move to MM.

Line 110: Move recipe of CDM to MM.

Lines 116-118: Where are growth rate values? Where they calculated. Add a graph with those data, calculation of growth rate important to make this point.

Line 149: It seems more fair to say that it's downregulated in the presence of Asn. Please reference Fig.1D.

Line 151: There is no statistics claiming the significance of the difference between +/- Asn in expression of GlnPQ. Add statistics to prove the difference is significant. The expression is just very marginally lower (even less than 2-fold lower). These are borderline differences and I am not convinced there is a real effect of Asn.

Line 153: You state that GlnPQ "...is an essential glutamine and glutamic acid uptake system and transports Asn in Gram-positive bacteria.", as already shown by Fulyani et al. Then you suggest that "...[it could] be responsible for Asn import." in GAS. Is this not stating the obvious? This was previously shown so it's not something new but a confirmation of what is already known. Please change your statement accordingly.

Line 157: The double mutant CovS and CovR is mentioned in the text but not present in Fig.1E and 1F, only a CovS mutant is present in the figure, please explain/amend?

Lines 158-159: Are cluster A and cluster B referring to the genes in Fig.1C and Fig.1D? Or where are these genes listed? Please make clear in the text also pointing at the correct panel in Fig.1. Also, in the legend of Fig.1 they are called Set1 and Set2 instead of cluster 1-2, the naming should be made consistent.

Lines 161-162: CovR and CovS deactivation is mentioned although only a covR mutant (S119 covR-) is present in Fig.EV1DE. Please specify if covR- is a double mutant (covR and CovS) or a single mutant (CovR) as it's not clear from the text. These results seem quite important since they show that CovR, but not CovS, play a role in Asn-mediated gene regulation. I would merge the graphs in Fig.1E-F with Fig.EVD-E.

Lines 166-167: "...we discovered that cluster B of genes encoding SLO, SLS, and GlnPQ..." this sentence is unclear. Genes present in cluster B?

Line 218: Growth rate is again mentioned but there are no data justifying the claim. Growth rates need to be calculated and

shown in a graph.

Line 250: Again growth rate mentioned but no values available.

Lines 251-252: The authors claim that transcription of cluster A genes is significantly reduced in the AsnB mutant. Compared to what? The expression level is the same as for the WT strain in the Absence of Asn. If at all, the gene expression does not vary compared to control conditions. Please rephrase. Also, there are no p-values on the graph to claim significance, those need to be added.

Lines 252-254: It is worth mentioning that the reduction of cluster B genes expression is independent on Asn addition to the medium, I would add this information

Line 253: The authors use the term "significantly reduced" although no p-values are indicated anywhere. Please add significance on the graphs

Line 287: Fig.EV1CD is wrongly cited, it should be Fig.EV1DE.

Line 291: It's incorrect to state that LL-37 stimulates the production of ScpC as no qPCR data are available. It's safe to say it leads instead to an increase in ScpC activity. Please modify this sentence accordingly

Lines 320-322: Mice cannot be attenuated, the severity of the presentation is, please modify accordingly.

Lines 350-352: The figure describing these findings is not cited in the text (presumably Fig.EVC?)

Line 353: I would specify the specific GAS strain tested. Be consistent throughout the manuscript since you describe the behavior of a few different GAS strains.

Line 366: Fig.6C is cited before Fig.6B, switch the figures or the labelling.

Lines 388-389: "Asn at both OD600 = 0.35 and 0.7, the ADP level of S119 388 grown in CDM is significantly higher than that grown in CDM only (Fig. 6D)". This sentence is not understandable and needs rephrasing.

Discussion:

As in the abstract, the discussion lacks a mention of the effect that the various mutants have on gene expression (basically half of the figures). Please add that and put it into context, adding the rationale of why you assessed gene expression and what your conclusions are.

Line 394: It "must derive" seems a bit strong in this context, since Asn can also be synthesized by GAS.

Lines 403-404 "We previously showed that extracellularly added Asn or formed in eukaryotic infected cells upregulates GAS virulence". This sentence needs restructuring for clarity reasons

Lines 404-406: Again, I am not convinced Asn metabolism regulates virulence, but rather Asn presence/absence. Please discuss extensively.

Lines 416-418: Why do you think AsnB is less virulent in mice? One would expect that with more Asn present the strain would be more virulent. Please discuss.

Lines 453-456: "We used Kidrolase successfully to treat intraperitoneal GAS infection in a mouse model, and Kidrolase prevented GAS growth in human blood, which serves as a culprit for GAS virulence." This sentence is very unclear, what "...serves as a culprit for GAS virulence"? Please reformulate.

Lines 467-468: Why did the authors not test treating the mice with one of such drugs to see if the infection severity was reduced for the WT strain? I think it would be an important control to have to prove their point. In case this was already done, please reference the findings in the appropriate place in the text.

Lines 470-473: The guidelines on how to treat GAS infections should be updated as these references are very old.

Methods:

Line 615: "Sublethal murine of human GAS soft tissue infection". What does that mean? Please reformulate

Figure legends:

Lines 874-875: There are no statistical values on the graphs of Fig.1B-C, please add them.

Lines 870-875: Unclear whether the fold change for dAsn+Asn are calculated relative to S119WT or to S119dAsn, specify in manuscript and figure legend.

Line 884: Again the genes examined are mentioned as Set1 and Set2 in the figure legend and as Cluster A-B in the text. Make it consistent.

Line 934: 50 genes is not "a few", "a few" would be max 3-4 genes.

Lines 957 and 964: Same comment as for line 884.

Lines 958-987: There is a verb missing in this sentence.

Lines 987-988: There is a "with" missing in this sentence.

Figures:

Fig.1A: Growth rate mentioned in the results section but no values calculated, missing data, please add.

Fig.1C: hylA and nga missing in this figure but present in Fig.1E, why?

Fig.1D: Is it really considered downregulation? Even less than 2 folds?

Fig.1C-D: These two panels are a repetition of the data presented in Fig.1E-F and Fig. EVD-E (S119+Asn). They are superfluous and I suggest cutting them.

Fig.2: Unclear whether the fold change of the S119dAsn +Asn is calculated based on S119WT or on S119dAsn. Please specify. Same comment applies for Fig.EV2B. in case everything is calculate relative to S119WT, a control S199dAsn without addition of Asn is missing in both figures. Please add the missing information.

Fig.3: Again statistical analysis is mentioned in the figure legend, but no p-values are indicated on the graphs. Please add the p-values to the graphs.

Fig.4: Please indicate the name of the specific GAS strain used in the figure legends, it is now only indicated as GAS.

Fig.4: Are all p-values indicated on the graphs? Please add if some are missing.

Fig.EV5 Panels D, E and F should be moved to the main figure as they are very nicely descriptive. I'd rather move spleen CFU counts to the supplemental material.

Referee #2:

The authors have responded to all comments of me and the reviewers, clarified a lot of technical detail, but I have the feeling the paper has not improved as much as it could, given the productive input provided by the referees. I guess my feeling is caused by the situation that the authors have fixed the detailed technical comments, but seem to have not taken the conceptual input on board. I don't want to block this paper- which I believe its not the job of a reviewer (its the paper of the authors not mine nor the journal's), and its not wrong either, just I feel a bit dissatisfied, seeing the potential the paper could have. I thus recommend to accept it, but my honest opinion is that the authors could have made more out of this one.

Point by point response to reviewer 1 (#2 previously)

Referee #1: Group A Streptococcal Asparagine Metabolism Regulates Bacterial
Virulence

The authors investigate the role of asparagine (Asn) in GAS virulence expression. They
could show that Asn improves GAS growth in minimal medium, enhances the
expression of several virulence factors (emm, ScpA, ScpC, Ska, HasA, Hyla, Nga), and
reduces the expression of the asparagine synthetase AsnA and the asparagine
transporter GlnP as well as of SLO and SLS, two streptolysins involved in scavenging
Asn from eukaryotic cells, in a negative feedback loop. The authors show that CovS is
required for Asn-mediated expression regulation of all genes examined, while CovR is
only necessary for Asn-mediated regulation of virulence factors but not for AsnA and
GlnP. They subsequently explore the role of AsnA and GlnP, both necessary for
expression regulation of all genes tested. The asparaginase AsnB was only necessary
for upregulation of virulence factors. They show that Asn increases virulence factors
activity by lowering intracellular ATP levels and therefore reducing CovR
phosphorylation. All genes involved in Asn metabolism (AsnA, GlnP and AsnB) are
necessary to enhance gas virulence in a murine necrotizing fasciitis mouse model.

**General comments:**

I am not sure the authors can claim that Asn metabolism regulates VFs in GAS, it
seems more regulated by the presence/absence of Asn, rather than its metabolism.
Please comment on that extensively (see also my comments below).

The summary description of the referee of our manuscript ignores the section entitled:
"Metabolism of Asn controls intracellular ADP/ATP levels." The uptake and synthesis of
Asn are ATP-consuming reactions and significantly impact the ATP/ADP ratio in GAS
because GAS uses glycolysis exclusively for energy generation. We quantified the
metabolites of strain S119 used in the study, which was grown in CDM in the presence
and absence of Asn at two OD₆₀₀ of 0.35 and 0.7. Specifically, we assessed the
metabolism of the intermediates of glycolysis, extracellular and intracellular Asn, ATP,
ADP, pyruvate, lactate, and others. We showed that the presence of Asn in the CDM
alters the GAS metabolism rate, and the RNAseq results also confirm these findings.
**Without performing the metabolic studies, we would not know that the availability**
**of extracellular Asn alters the metabolism and thus the ADP/ATP ratio, which**
**controls the dephosphorylation level of CovRS, which is the main finding of this**
**manuscript.** The fact that mutants in AsnA, AsnB, and GlnPQ have altered metabolism
compared to that of the WT, had higher ATP/ADP ratios, and were all attenuated in a
murine model of GAS human infection strongly supports that the metabolic changes we
observed in the minimal CDM are relevant to the *in vivo* GAS infection. Thus, we believe
that the title of this manuscript is appropriate.

Point by point response to reviewer 1 (#2 previously)

The manuscript quality is partly poorly written and should be made more concise for
clarity. The text is embedded with many imprecisions, both graphical and grammatical,
such as figures incorrectly labeled, statistics on figures missing, and sentences missing
subject/verb, just to mention some. The titles in the result section should be more
explanatory, describing the main finding of the paragraph and reflecting the title of the
figure legends.

The manuscript was read by Kevin McIver (Co-author), a native English speaker/writer
who did not point to significant problems with the writing style. Before sending
manuscripts (more than 30 years of practice), I ran them routinely through the "Editor"
and "Grammarly" programs, and in the case of this manuscript, I did not see any
mistakes in spelling or grammar. We are confident that the linguistic editors of EMBOR
can help deal with residual cavities in the style if they exist. The abstract, some of the
titles of the paragraphs, and the figure legends were changed according to the
reviewer's suggestions, and the statistical P-values are shown in the figures as required.

**Abstract:**

There is no mention of Asn-mediated gene expression regulation in the abstract, which I
is interesting since these data cover about half of the findings. This information needs to
be embedded in the abstract in a seamless way. A summary of the manuscript may be
better described at the end of the introduction, for instance (line 89-103).

Since this work is complex and includes results based on a combination of several key
independent complicated methodologies performed ex-vivo, in-vivo, and in-vitro, it is not
easy to cover all the new findings seamlessly in abstract (and within the word limit), as
the reviewer wishes. Nevertheless, we altered the abstract and hope the updated
version satisfies the reviewer.

Line 31: Human manifestations? I'd change to human diseases.

The text was changed to human disease (line 30).

Lines 32-35: "Here, we show that asparagine (Asn) homeostasis in GAS is maintained
by asparagine synthetase (AsnA), the ABC transporter (GlnPQ) that imports
extracellular Asn, and a highly efficient asparaginase (AsnB) that degrades intracellular
Asn". This is presented as a main finding in the abstract but this does not reflect what's
in the manuscript. Additionally, I also fear the authors can't claim that they discover how

Point by point response to reviewer 1 (#2 previously)

Asn homeostasis is maintained in GAS since the function of all genes they cite is
already known and they are all already known to be involved in Asn homeostasis.
Please comment on this and change the abstract accordingly.

From any genome sequence of each of the almost 4000 genomes of GAS sequenced,
one can see that GAS possesses a single copy of a gene encoding AsnA and AsnB, so
if this gene products are expressed under the conditions used, GAS must produce Asn
from aspartate and ammonia by AsnA and degrades it to back aspartate and ammonia
by AsnB. Is this information sufficient to conclude how Asn homeostasis is achieved in
GAS? The answer is no.

1. The transporter responsible for the uptake of Asn from the medium or the host
was unknown, and in this manuscript, we provide the first **unambiguous**
**experimental evidence** that the ABC transporter GlnPQ is the primary
transporter responsible for the uptake of Asn in GAS. Indeed, GlnPQ serves as
an uptake system for amino acids in Gram-positive pathogens, including
glutamine, glutamate, and Asn, but one needs to prove that GlnPQ is exclusively
responsible for Asn import. For example, GlnPQ is the exclusive glutamine
transporter in *Listeria*, which has been proven experimentally (Haber *et al.*,
2017). The GlnPQ transporter has two tandem substrate-binding domains
(SBDs) fused to each transmembrane domain, giving rise to four SBDs per
functional transporter. Each SBD can bind more than one amino acid at high
affinity. SBD1 binds Asn and arginine, and SBD2 binds glutamine and glutamate
(Nemchinova *et al.*, 2024). So, without proving it experimentally, it is dubious to
state that GlnPQ is the primary transporter for Asn in GAS, which is evident from
the literature.

Our past studies showed that the transcription of several ABC
transporters, including *glnPQ* (see attached table), were downregulated in strain
JS95 (M14) by the absence of Asn, caused by the addition of Kidrolase to a
semi-rich GAS medium (see attached table). Here, we demonstrated that *glnPQ*
transcription is upregulated by the absence and downregulated by the presence
of Asn S119 grown in CDM. These pieces of information are insufficient to prove
that GlnPQ is the primary transporter in GAS. An experimental approach is
required.

**In conclusion, one cannot determine whether GlnPQ is the primary**
**ABC transporter for Asn in GAS a priori. This point must be proven**
**experimentally by constructing an appropriate inactive mutant and**
**measuring Asn transport as we did in this manuscript.**

Table for referee with unpublished data and its description has been removed upon request by the authors.

2. The regulation modes of the genes encoding AsnA and AsnB by Asn have not been studied before in GAS. We show that Asn downregulates the expression of AsnA. Moreover, the AsnA mutant shows a significant increase in the expression of *glnPQ* (about 6-fold Fig. 2C). So, the Asn homeostasis in GAS is far from being as apparent as the reviewer implies and occurring both at post-translational and translational levels during GAS growth and host infection and includes at least three players: AsnA, GlnPQ, AsnB.
3. AsnB belongs to the highly efficient class II asparaginase. Indeed, we show that a mutant in AsnB produces elevated levels of intracellular Asn compared to WT. One could suggest that the mutant would be highly virulent since Asn promotes the expression of virulence genes. However, this is not the case. This mutant does not produce an elevated level of virulence and is attenuated in the mouse model of human NF. We assume that an excess of Asn alters GAS metabolism to produce more ATP because it down-regulates Asn uptake and formation, both ATP-consuming reactions. Thus, our manuscript shows how homeostasis of Asn is supported and how keeping intracellular Asn within a range of concentrations is important for regulating virulence due to alteration of the ADP/ATP ratio.

Lines 92-93 state, "Furthermore, we found that the ABC transporter GlnPQ is responsible for importing Asn...". Is this not already known from the literature? Please comment!

The answer to the reviewer's question is, again, no. The repeated reviewer's questions are equivalent to asking the following: A homolog to CovR/S is present in Group B streptococcus. Thus, the GAS mode of gene regulation by CovR/S must

Point by point response to reviewer 1 (#2 previously)

be known since the regulation by CovR/S GBS has been extensively studied. Isn't
it? The answer is obvious.

There is no description of the mechanisms of Asn-mediated tumor growth in the
introduction that needs to be added. The authors suggest a similarity but do not
describe what happens in tumor cells.

The mechanism summary appears in lines 435-451, where this point is discussed in
detail.

Line 55: Bacterial infections or generally?

We do not understand this comment.

Line 66: It was repeatedly shown that GAS is mainly an extracellular pathogen but can
also survive intracellularly, this has to be specified here.

The role of the temporary intracellular existence of GAS is a controversial issue. We
were among the first labs to show that GAS can penetrate eukaryotic cells. J Infect Dis.
1998 Jul;178(1):147-58.doi: 10.1086/515589. Since then, it has been shown that a
small population of GAS cells can temporarily live within eukaryotic cells, but this entry's
physiological role and relevance to *in-vivo* infection is unknown. Therefore, we do not
feel this phenomenon is relevant to our study and should be specified or cited in our
manuscript. In our experience, many studies performed with eukaryotic cells infected
with MOI exceeding 1 (which allows GAS penetration into eukaryotic cells) are probably
irrelevant to GAS pathogenesis or perhaps occur at the infection's last stages, just
before the host and the bacterial death.

Lines 92-93: "Furthermore, we found that the ABC transporter GlnPQ is responsible for
importing Asn...". Is this not already known from the literature? Please comment!

We addressed this point; please see above.

**Results:**

Point by point response to reviewer 1 (#2 previously)

As mentioned, the titles in the result sections should be more explanatory, describing
the main finding of the paragraph. Please change them accordingly!

**Some of the titles were altered.**

Lines 107-108: Move to MM.

**The lines were moved.**

Line 110: Move recipe of CDM to MM.

**We have moved the CDM reference to MM.**

Lines 116-118: Where are growth rate values? Where they calculated. Add a graph with
those data, calculation of growth rate important to make this point.

**We have already reported Asn's-mediated acceleration of GAS growth in:**

1. **An extracellular bacterial pathogen modulates host metabolism to regulate its**
**own sensing and proliferation. Baruch M, Belotserkovsky I, Hertzog BB, Ravins**
**M, Dov E, McIver KS, Le Breton YS, Zhou Y, Cheng CY, Hanski E. Cell. 2014 Jan**
**16;156(1-2):97-108. doi: 10.1016/j.cell.2013.12.007.**

2. **Unfolded protein response inhibitors cure group A streptococcal necrotizing**
**fasciitis by modulating host asparagine. Anand A, Sharma A, Ravins M, Biswas**
**D, Ambalavanan P, Lim KXZ, Tan RYM, Johri AK, Tirosh B, Hanski E. Sci Transl**
**Med. 2021 Aug 4;13(605):eabd7465. doi: 10.1126/scitranslmed.abd7465.**

**Here, we changed the term "growth rate" to growth kinetics. The changes in growth**
**kinetics are apparent and provided mainly to confirm the phenotypes of the indicated**
**genetic manipulations (except in Fig. 1A). The increase in growth kinetics is self-**
**evident in Fig. 1A from the growth curve as a function of time at the initial stages in**
**which OD and time show a linear dependency. As indicated above, this phenomenon**
**has been reported and published already twice. Therefore, we strongly feel that**
**calculations of growth rates are not needed for this study.**

Line 153: You state that GlnPQ "...is an essential glutamine and glutamic acid uptake
system and transports Asn in Gram-positive bacteria.", as already shown by Fulyani et
al. Then you suggest that "...[it could] be responsible for Asn import." in GAS. Is this not

Point by point response to reviewer 1 (#2 previously)

stating the obvious? This was previously shown so it's not something new but a
confirmation of what is already known. Please change your statement accordingly.

This issue is discussed at length above. We changed the text to clarify the point (lines
76-87).

Line 157: The double mutant CovS and CovR is mentioned in the text but not present in
Fig.1E and 1F, only a CovS mutant is present in the figure, please explain/amend?

As the reviewer pointed out, the CovRS is an operon. Thus, an insertional mutation in
CovR creates deletion of both genes, and ΔcovS is a precise deletion, resulting in a
mutant expressing *covR*. Both mutants are described in detail in MM. The results of
ΔcovS are presented in Fig. 1, E, and F, while for CovR-, in the supplementary. We
agree with the reviewer that the CovR- results should be transferred to the main text as
suggested below, and the composition of the mutants was explained better in the text.
(see lines 157-168).

As the Lines 158-159: Are cluster A and cluster B referring to the genes in Fig.1C and
Fig.1D? Or where are these genes listed? Please make clear in the text also pointing at
the correct panel in Fig.1. Also, in the legend of Fig.1 they are called Set1 and Set2
instead of cluster 1-2, the naming should be made consistent.

We thank the reviewer for pointing out this inconsistency and have changed the text
accordingly to Set 1 and Set 2.

Lines 161-162: CovR and CovS deactivation is mentioned although only a *covR* mutant
(S119 *covR*-) is present in Fig.EV1DE. Please specify if *covR*- is a double mutant (*covR*
and CovS) or a single mutant (CovR) as it's not clear from the text. These results seem
quite important since they show that CovR, but not CovS, play a role in Asn-mediated
gene regulation. I would merge the graphs in Fig.1E-F with Fig.EVD-E.

We agree with the reviewer and merged the graphs. As described above, we clarified
this point in the text (lines 157-168). (The construction of the mutants is described in
detail in MM).

Point by point response to reviewer 1 (#2 previously)

Lines 166-167: "...we discovered that cluster B of genes encoding SLO, SLS, and
GlnPQ..." this sentence is unclear. Genes present in cluster B?

We clarified the text (please see lines 168-171)

Line 218: Growth rate is again mentioned but there are no data justifying the claim.
Growth rates need to be calculated and shown in a graph.

Please see our answer above.

Line 250: Again growth rate mentioned but no values available.

Please see our answer above.

Lines 251-252: The authors claim that transcription of cluster A genes is significantly
reduced in the AsnB mutant. Compared to what? The expression level is the same as
for the WT strain in the Absence of Asn. If at all, the gene expression does not vary
compared to control conditions. Please rephrase. Also, there are no p-values on the
graph to claim significance, those need to be added.

We rephrased the text to make it more understood to the reader (lines 244-248). The
comparison is, of course, to that of S119 grown in CDM with Asn. The P-values were
added and are shown on the graph.

Lines 252-254: It is worth mentioning that the reduction of cluster B genes expression is
independent on Asn addition to the medium, I would add this information.

We agree with the reviewer and added the information (lines 247-248)

Line 253: The authors use the term "significantly reduced" although no p-values are
indicated anywhere. Please add significance on the graphs

The P-values were added to all the figures.

Point by point response to reviewer 1 (#2 previously)

Line 287: Fig.EV1CD is wrongly cited, it should be Fig.EV1DE.

**As suggested by the reviewer, the figures EV1D,E were moved to the main figures part**
**and are numbered now as (Fig. 1G,H).**

Line 291: It's incorrect to state that LL-37 stimulates the production of ScpC as no qPCR
data are available. It's safe to say it leads instead to an increase in ScpC activity. Please
modify this sentence accordingly

**We changed the text- to an increase in ScpC activity.**

Lines 320-322: Mice cannot be attenuated, the severity of the presentation is, please
modify accordingly.

**The reviewer is correct. We changed the text accordingly (lines 316-318).**

Lines 350-352: The figure describing these findings is not cited in the text (presumably
Fig.EVC?)

**The figure has been cited correctly in the text as Fig. EV6C (line 351)**

Line 353: I would specify the specific GAS strain tested. Be consistent throughout the
manuscript since you describe the behavior of a few different GAS strains.

**We do not understand the comment -all the different strains were specified in the figure.**

Line 366: Fig.6C is cited before Fig.6B, switch the figures or the labelling.

**We corrected the positions of the figures as suggested.**

Lines 388-389: "Asn at both OD600 = 0.35 and 0.7, the ADP level of S119 388 grown in
CDM is significantly higher than that grown in CDM only (Fig. 6D)". This sentence is not
understandable and needs rephrasing.

**We rephrased it in the text.**

Point by point response to reviewer 1 (#2 previously)

Discussion:

As in the abstract, the discussion lacks a mention of the effect that the various mutants
have on gene expression (basically half of the figures). Please add that and put it into
context, adding the rationale of why you assessed gene expression and what your
conclusions are.

The text of the discussion was altered. The regulation appeared in the original
discussion lines (419-428); we expanded it.

Line 394: It "must derive" seems a bit strong in this contest, since Asn can also be
synthesized by GAS.

GAS is multiple-amino-acid-auxotrophic (see Davies *et al.*, 1965, old but still valid), as
provided in the text.

Lines 403-404 "We previously showed that extracellularly added Asn or formed in
eukaryotic infected cells upregulates GAS virulence". This sentence needs restructuring
for clarity reasons.

We have rephrased the sentence.

Lines 404-406: Again, I am not convinced Asn metabolism regulates virulence, but
rather Asn presence/absence. Please discuss extensively.

It was discussed above comprehensively.

Lines 416-418: Why do you think AsnB is less virulent in mice? One would expect that
with more Asn present the strain would be more virulent. Please discuss.

We added the probable reason to the explanation (please see lines 121-131 here in P-
B-P response).

Lines 453-456: "We used Kidrolase successfully to treat intraperitoneal GAS infection in
a mouse model, and Kidrolase prevented GAS growth in human blood, which serves as
a culprit for GAS virulence." This sentence is very unclear, what "...serves as a culprit
for GAS virulence"? Please reformulate.

The sentence was reformulated.

Point by point response to reviewer 1 (#2 previously)

Lines 467-468: Why did the authors not test treating the mice with one of such drugs to
see if the infection severity was reduced for the WT strain? I think it would be an
important control to have to prove their point. In case this was already done, please
reference the findings in the appropriate place in the text.

1. This experiment was published for JS95 M14 strain: Unfolded protein response
inhibitors cure group A streptococcal necrotizing fasciitis by modulating host
asparagine. Anand A, Sharma A, Ravins M, Biswas D, Ambalavanan P, Lim KXZ,
Tan RYM, Johri AK, Tirosh B, Hanski E. *Sci Transl Med*. 2021 Aug
4;13(605):eabd7465. doi: 10.1126/scitranslmed.abd7465.

It also is true for strain S119 (see attached figure). We do not think it serves as an
important control for this manuscript. The mechanism of scavenging Asn is shared by
many strains we examined, including S119.

*Figure for referee with unpublished data and its description has been removed upon*
request by the authors.

Lines 470-473: The guidelines on how to treat GAS infections should be updated as
these references are very old. Methods:

We added one reference from 2024. Nevertheless, there is no significant change in the
treatment.

Line 615: "Sublethal murine of human GAS soft tissue infection". What does that mean?
Please reformulate.

Point by point response to reviewer 1 (#2 previously)

We rephrased the sentence.

Figure legends:

Lines 874-875: There are no statistical values on the graphs of Fig.1B-C, please add
them.

The P-values were added.

Lines 870-875: Unclear whether the fold change for dAsn+Asn are calculated relative to
S119WT or to S119dAsn, specify in manuscript and figure legend.

We used S119 WT without Asn as a control for all qPCR. It has been specified in the
manuscript and figure legends.

Line 884: Again the genes examined are mentioned as Set1 and Set2 in the figure
legend and as Cluster A-B in the text. Make it consistent.

We have corrected it.

Line 934: 50 genes is not "a few", "a few" would be max 3-4 genes.

We have corrected it. (line 922)

Lines 957 and 964: Same comment as for line 884.

We have corrected it to Set 1 and Set 2.

Lines 958-987: There is a verb missing in this sentence. Lines 987-988: There is a
"with" missing in this sentence.

We have corrected it.

Figures:

Fig.1A: Growth rate mentioned in the results section but no values calculated, missing
data, please add.

We addressed this above in detail.

Point by point response to reviewer 1 (#2 previously)

Fig.1C: hylA and nga missing in this figure but present in Fig.1E, why?

**This point was already addressed in detail in the previously revised version.**

Fig.1D: Is it really considered downregulation? Even less than 2 folds?

**The p-values provided show significant downregulation.**

Fig.1C-D: These two panels are a repetition of the data presented in Fig.1E-F and Fig.
EVD-E (S119+Asn). They are superfluous and I suggest cutting them. Fig.2: Unclear
whether the fold change of the S119dAsn +Asn is calculated based on S119WT or on
S119dAsn.

**This point was already modified in Figure Legends.**

Please specify. Same comment applies for Fig.EV2B. in case everything is calculate
relative to S119WT, a control S199dAsn without addition of Asn is missing in both
figures. Please add the missing information.

**As already answered in the earlier revised version delta AsnA is an auxotroph.**

Fig.3: Again statistical analysis is mentioned in the figure legend, but no p-values are
indicated on the graphs. Please add the p-values to the graphs. Fig.4:

**The P values were added.**

Please indicate the name of the specific GAS strain used in the figure legends, it is now
only indicated as GAS. Fig.4: Are all p-values indicated on the graphs? Please add if
some are missing.

**It is specified in the figure.**

Fig.EV5 Panels D, E and F should be moved to the main figure as they are very nicely
descriptive. I'd rather move spleen CFU counts to the supplemental material.

**We have done it as suggested.**

Prof. Emanuel Hanski
The Hebrew University Faculty of Medicine
Ein Karem Campus
Jerusalem 91102
Israel

Dear Prof. Hanski,

Thank you for the submission of your further revised manuscript to our editorial offices. I now looked through the revised manuscript and your p-b-p-response and consider the remaining points of the referee as adequately addressed. I am thus very pleased to accept your manuscript for publication in the next available issue of EMBO reports. Thank you for your contribution to our journal.

Yours sincerely,
